

# Earth System Model Aerosol-Cloud Diagnostics Package (ESMAC Diags) Version 2: Assessments of Aerosols, Clouds and Aerosol-Cloud Interactions Through Field Campaign and Long-Term Observations

Shuaiqi Tang[1], Adam C. Varble[1], Jerome D. Fast[1], Kai Zhang[1], Peng Wu[1], Xiquan Dong[2], Fan Mei[1], Mikhail Pekour[1], Joseph C. Hardin[3], and Po-Lun Ma[1]

[1]Pacific Northwest National Laboratory, Richland, WA, USA
[2]University of Arizona, Tucson, AZ, USA
[3]Unaffiliated scientist

*Correspondence to*: Shuaiqi Tang (shuaiqi.tang@pnnl.gov)

**Abstract.**

Poor representations of aerosols, clouds and aerosol-cloud interactions (ACI) in Earth System Models (ESMs) have long been the largest uncertainties in predicting global climate change. Huge efforts have been made to improve the representation of these processes in ESMs, and key to these efforts is evaluation of ESM simulations with observations. Most well-established ESM diagnostics packages focus on the climatological features; however, they are lack of the process-level understanding and representations of aerosols, clouds, and ACI. In this study, we developed an ESM aerosol-cloud diagnostics package (ESMAC Diags) to facilitate routine evaluation of aerosols, clouds and aerosol-cloud interactions simulated by the Department of Energy's (DOE) Energy Exascale Earth System Model (E3SM). This paper documents its version 2 functionality (ESMAC Diags v2), which has substantial updates from its version 1 (Tang et al., 2022a). The simulated aerosol and cloud properties have been extensively compared with in-situ and remote-sensing measurements from aircraft, ship, surface and satellite platforms in ESMAC Diags v2. It currently includes six field campaigns and two permanent sites covering four geographical regions: Eastern North Atlantic, Central U.S., Northeastern Pacific and Southern Ocean, where frequent liquid or mixed-phase clouds are present and extensive measurements are available from the DOE Atmospheric Radiation Measurement user facility and other agencies. ESMAC Diags v2 generates various types of single-variable and multi-variable diagnostics, including percentiles, histograms, joint histograms and heatmaps, to evaluate model representation of aerosols, clouds, and aerosol-cloud interactions. Select examples highlighting ESMAC Diags capabilities are shown using E3SM version 2 (E3SMv2). E3SMv2 in general can reasonably reproduces many observed aerosol and cloud properties, with biases in some variables such as aerosol particle and cloud droplet sizes and number concentrations. The coupling of aerosol and cloud number concentrations may be too strong in E3SMv2, possibly indicating a bias in processes that control aerosol activation. Furthermore, the liquid water path adjustment to perturbed cloud droplet number concentration behaves differently in E3SMv2 and observations, which warrants a further study to improve the cloud microphysics parameterizations in E3SMv2.



## 1. Introduction

Poor representations of aerosols, clouds and aerosol-cloud interactions (ACI) in Earth System Models
(ESMs) have long been the largest uncertainties in predicting global climate change (IPCC, 2021).
Challenges come from several aspects: first, there are many aerosol properties (e.g., number, size, phase,
shape, composition) and cloud micro- and macro-physical properties (e.g., fraction, water content,
number and size of liquid and ice hydrometeors) that affect Earth's climate. Coincident measurements of
these properties remain largely under-sampled due to substantial spatiotemporal variability and logistical
difficulties for making such measurements. Second, there are complex interactive processes between
aerosols, clouds, and ambient meteorological conditions, many of which are not fully understood, but are
critical to properly interpreting relationships between observable properties. Third, many ACI processes
are nonlinear, multi-scale processes that involve feedbacks depending on cloud types and meteorological
regimes, which also shift in space and time, presenting challenges for assessing causal effect and
representing such processes in ESMs.
Huge efforts have been made to improve the representation of aerosols, clouds and ACI in ESMs. Key to
these efforts is evaluation of ESM simulations with observations. Many modeling centers have developed
standardized diagnostics packages to document ESM performance. For aerosol and cloud properties, most
diagnostic packages rely heavily on satellite measurements as evaluation data (e.g., AMWG, 2021;
E3SM, 2021; Eyring et al., 2016; Gleckler et al., 2016; Maloney et al., 2019; Myhre et al., 2013; Schulz
et al., 2006). Satellite remote sensing measurements have global or near global coverage but limited
spatial and temporal resolution. They are also unable to retrieve some variables, especially for aerosol
properties such as cloud condensation nuclei (CCN) number concentration, while many cloud
microphysical retrievals such as droplet number concentration have large uncertainties (e.g., Grosvenor et
al., 2018). This limits their application to robustly quantify aerosols, clouds and ACI processes. In-situ
measurements from ground, aircraft or ship platforms from field campaigns are also used in a few
projects to evaluate ESMs (e.g., Reddington et al., 2017; Watson-Parris et al., 2019; Tang et al., 2022a;
Zhang et al., 2020). Some of these field campaigns were conducted over remote or poorly sampled
locations, which are highly valuable for model evaluation despite limited spatial coverage and time
periods. Moreover, the DOE Atmospheric Radiation Measurement (ARM) user facility has conducted
continuous field measurements at a few sites for multiple years. These long-term high-resolution field
measurements have also been demonstrated to be valuable for evaluating ESMs (e.g., Zhang et al., 2020).
In response to the need for more ESM diagnostics for evaluating ACI processes, Tang et al. (2022a)
developed an ESM aerosol-cloud diagnostics package (ESMAC Diags) to facilitate the routine evaluation
of aerosols, clouds and ACI simulated by the Department of Energy's (DOE) Energy Exascale Earth
System Model (E3SM, Golaz et al., 2019). It includes diagnostics that leverage in-situ measurements
from multiple platforms during six field campaigns since 2013, which are not included in previous
diagnostics tools (e.g., Reddington et al., 2017). Version 1 of ESMAC Diags (ESMAC Diags v1, Tang et
al., 2022a) mainly focuses on aerosol properties. We present here version 2 of ESMAC Diags (ESMAC
Diags v2) that is a direct extension of ESMAC Diags v1 with two major additions:

77        1. measurements from satellite and long-term diagnostics at the ARM Southern Great Plains
(SGP) and Eastern North Atlantic (ENA) sites.



2. diagnostics for cloud properties and aerosol-cloud interactions.
The new measurements, as well as major data quality controls are introduced in Section 2. Additional
discussions on retrieval uncertainties of cloud microphysical properties are performed in Section 3.
Details of the code structure of ESMAC Diags v2, which is substantially changed since version 1, are
described in Section 4. Section 5 provides selected examples of single-variable and multi-variable
diagnostics using ESMAC Diags v2 to highlight its capabilities. Lastly, Section 6 provides a summary.
**2. Aerosol and cloud measurements from ground, aircraft, ship and satellite platforms**
Following the initial development in version 1, ESMAC Diags v2 continues to focus on six field
campaigns conducted in four geographical regions: the Central U.S. (CUS, where the ARM Southern
Great Plains (SGP) site is located), Eastern North Atlantic (ENA), Northeastern Pacific (NEP), and
Southern Ocean (SO). Information on the six field campaigns is shown in Table 1 and their locations are
shown in Figure 1, each reproduced from Table 1 and Figure 3 in Tang et al. (2022a).

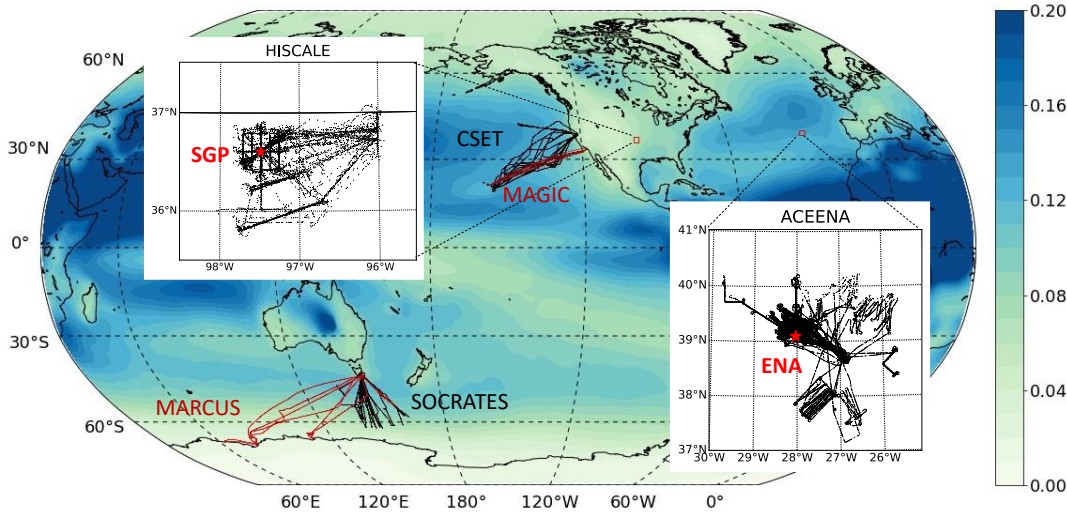


Figure 1. Aircraft (black) and ship (red) tracks for the six field campaigns. Red stars in the
enlarged map indicate two ARM fixed sites: SGP and ENA, that have long-term
measurements available for model diagnostics. Overlaid is aerosol optical depth at
550nm averaged from 2014 to 2018 simulated in E3SMv1. (Reproduced from Figure 3 in
Tang et al., 2022a)
Table 1. Descriptions of the field campaigns used in this study. (Reproduced from Table 1
in Tang et al., 2022a)

| Campaign* | Period | Platform | Typical Conditions | Reference |
|---|---|---|---|---|
| **HI-SCALE** | IOP1: 24 Apr – 21 May 2016 IOP2: 28 Aug – 24 Sep 2016 | Ground, aircraft (IOP1: 17 flights, IOP2: 21 flights) | Continental cumulus with high aerosol loading | (Fast et al., 2019) |



| ACE-ENA | IOP1: 21 Jun – 20 Jul 2017 IOP2: 15 Jan – 18 Feb 2018 | Ground, aircraft (IOP1: 20 flights, IOP2: 19 flights) | Marine stratocumulus with low aerosol loading | (Wang et al., 2021) |
|---|---|---|---|---|
| MAGIC | Oct 2012 – Sep 2013 | Ship (18 legs) | Marine stratocumulus to cumulus transition with low aerosol loading | (Lewis and Teixeira, 2015; Zhou et al., 2015) |
| CSET | 1 Jul – 15 Aug 2015 | Aircraft (16 flights) | Same as above | (Albrecht et al., 2019) |
| MARCUS | Oct 2017 – Apr 2018 | Ship (4 legs) | Marine liquid and mixed phase clouds with low aerosol loading | (McFarquhar et al., 2021) |
| SOCRATES | 15 Jan – 24 Feb, 2018 | Aircraft (14 flights) | Same as above | (McFarquhar et al., 2021) |

* Full names of the listed field campaigns:
HI-SCALE: Holistic Interactions of Shallow Clouds, Aerosols and Land Ecosystems
ACE-ENA: Aerosol and Cloud Experiments in the Eastern North Atlantic
MAGIC: Marine ARM GCSS Pacific Cross-section Intercomparison (GPCI) Investigation of Clouds
CSET: Cloud System Evolution in the Trades
MARCUS: Measurements of Aerosols, Radiation and Clouds over the Southern Ocean
SOCRATES: Southern Ocean Cloud Radiation and Aerosol Transport Experimental Study

The collection and processing of observations are the most time-consuming part of developing ESMAC
Diags, which also impacts the reliability of conclusions drawn from the model diagnostics. In this section,
we introduce the data used in ESMAC Diags v2, existing quality issues in some datasets, and treatments
to address these quality issues. Some variables are difficult to directly measure or have limited in-situ
sampling and thus must be derived from remote sensing measurements using retrieval algorithms. In
Section 3, we further discuss the uncertainty and reliability of some cloud retrieval products via
comparisons with in-situ aircraft measurements.
2.1. Data availability
All measurements, instruments, and data products used in the six field campaigns and two long-term sites
in ESMAC Diags v2 are shown in Table 2. Further details of the measurements, data product names, and
DOIs are given in Tables S1 to S6 (for field campaigns) and Tables S7 and S8 (for SGP and ENA sites) in
the supplementary material. To allow maximum overlapping of key measurements while also ensuring a
long enough period for statistical evaluation, we select the periods of 1 Jan 2011 – 31 Dec 2020 for SGP
and 1 Jan 2016 – 31 Dec 2018 for ENA for long-term analyses. In addition to the aerosol measurements
discussed in Tang et al. (2022a), we incorporate more cloud and radiation measurements, as well as
geostationary satellite retrievals using Visible Infrared Solar-Infrared Split Window Technique (VISST)
(Minnis et al., 2008; Minnis et al., 2011) algorithm. The VISST products archived by ARM cover
approximately 10° by 10° regions in 0.5° by 0.5° resolution centered over ARM sites. Moreover, ARM
recently released products consisting of merged aerosol particle and cloud droplet size distributions from
aircraft measurements for HI-SCALE and ACE-ENA campaigns. These data are now used in ESMAC
Diags v2.
Table 2: List of instruments and measurements used in ESMAC Diags v2.





| Platform | Measurements | Instruments / data products | Available campaigns |
|---|---|---|---|
| **Ground** | Surface temperature, relative humidity, wind, pressure, precipitation; upper-level temperature, relative humidity, wind | Surface meteorological station (MET), ARM best estimate (ARMBE) products | HI-SCALE, ACE-ENA, SGP, ENA |
| | Longwave and shortwave radiation, cloud fraction | ARM best estimate (ARMBE) products | HI-SCALE, ACE-ENA, SGP, ENA |
| | Aerosol number concentration | Condensation particle counter (CPC), Condensation particle counter – fine (CPCF), Condensation particle counter – ultrafine (CPCU), Ultra-high sensitivity aerosol spectrometer (UHSAS), Scanning mobility particle sizer (SMPS) | HI-SCALE, ACE-ENA, SGP, ENA |
| | Aerosol size distribution | Ultra-high sensitivity aerosol spectrometer (UHSAS), Scanning mobility particle sizer (SMPS), Nano scanning mobility particle sizer (nanoSMPS) | HI-SCALE, ACE-ENA, SGP, ENA |
| | Aerosol composition | Aerosol chemical speciation monitor (ACSM) | HI-SCALE, ACE-ENA, SGP, ENA |
| | CCN number concentration | Cloud condensation nuclei (CCN) counter | HI-SCALE, ACE-ENA, SGP, ENA |
| | Cloud optical depth | Multifilter rotating shadowband radiometer (MFRSR) | HI-SCALE, ACE-ENA, SGP, ENA |
| | Cloud droplet number concentration | Cloud droplet number concentration retrieval (Ndrop), cloud retrieval from Wu et al. (2020) | HI-SCALE, ACE-ENA, SGP, ENA |
| | Cloud droplet effective radius | Multifilter rotating shadowband radiometer (MFRSR), cloud retrieval from Wu et al. (2020) | HI-SCALE, ACE-ENA, SGP, ENA |
| | Cloud liquid water path | Microwave radiometer (MWR), ARM best estimate (ARMBE) products | HI-SCALE, ACE-ENA, SGP, ENA |
| | Cloud base height, cloud top height | Active remote sensing of clouds (ARSCL) | HI-SCALE, ACE-ENA, SGP, ENA |
| **Satellite** | TOA shortwave and longwave radiation | Geostationary satellite-based retrievals using Visible Infrared Solar-Infrared Split Window Technique (VISST) algorithm | HI-SCALE, ACE-ENA, MAGIC, MARCUS, SGP, ENA |
| | cloud fraction; height, pressure and temperature at cloud top | Geostationary satellite-based retrievals using Visible Infrared Solar-Infrared Split Window Technique (VISST) algorithm | HI-SCALE, ACE-ENA, MAGIC, MARCUS, SGP, ENA |
| | liquid water path; cloud optical depth; droplet effective radius | Geostationary satellite-based retrievals using Visible Infrared Solar-Infrared Split Window Technique (VISST) algorithm | HI-SCALE, ACE-ENA, MAGIC, MARCUS, SGP, ENA |
| | Cloud droplet number concentration | Retrieved from VISST data using the algorithm in Bennartz (2007) | HI-SCALE, ACE-ENA, MAGIC, MARCUS, SGP, ENA |
| **Aircraft** | Navigation information and meteorological parameters | Interagency working group for airborne data and telemetry systems (IWG) | HI-SCALE, ACE-ENA |
| | Aerosol number concentration | Condensation particle counter (CPC), Condensation particle counter – ultrafine (CPCU), Condensation nuclei counter (CNC), Ultra-high sensitivity aerosol spectrometer (UHSAS), Passive cavity aerosol spectrometer (PCASP) | HI-SCALE, ACE-ENA, CSET, SOCRATES |
| | Aerosol size distribution | Ultra-high sensitivity aerosol spectrometer (UHSAS), Fast integrated mobility spectrometer (FIMS), Passive cavity aerosol spectrometer (PCASP), Best estimate aerosol size distribution (BEASD) | HI-SCALE, ACE-ENA, CSET, SOCRATES |
| | Aerosol composition | High-resolution time-of-flight aerosol mass spectrometer (AMS) | HI-SCALE, ACE-ENA |





| | CCN number concentration | Cloud condensation nuclei (CCN) counter | HI-SCALE, ACE-ENA, SOCRATES |
|---|---|---|---|
| | Cloud liquid water content | Water content measuring system (WCM), PMS-King Liquid Water Content (LWC) | HI-SCALE, ACE-ENA, CSET, SOCRATES |
| | Cloud droplet number size distribution | 1DC, 2DC, 2DS, CDP, Cloud probe merged size distribution (mergedSD) | HI-SCALE, ACE-ENA, CSET, SOCRATES |
| **Ship** | Navigation information and meteorological parameters | Meteorological station (MET) | MAGIC, MARCUS |
| | Aerosol number concentration | Condensation particle counter (CPC), Ultra-high sensitivity aerosol spectrometer (UHSAS) | MAGIC, MARCUS |
| | Aerosol size distribution | Ultra-high sensitivity aerosol spectrometer (UHSAS) | MAGIC, MARCUS |
| | CCN number concentration | Cloud condensation nuclei (CCN) counter | MAGIC, MARCUS |
| | Cloud liquid water path | Microwave radiometer (MWR) | MAGIC, MARCUS |
| | Cloud droplet number concentration, cloud effective radius | Cloud retrieval from Wu et al. (2020) | MAGIC |


All the observational data are quality controlled with their time resolution re-scaled to that suitable for
evaluating E3SM. Currently, ground, ship and satellite measurements are re-scaled to a 1-hour frequency
which is approximately consistent with 1-degree resolution E3SM output. Rescaling consists of
computing either the median, mean or interpolated value depending on the original data frequency and
variable properties. For most aerosol and cloud microphysics measurements, the median value is
computed to remove occasional spikes or zeros resulting from data contamination or measurement error.
For some bulk cloud properties (e.g., cloud fraction, liquid water path (LWP)), the mean value is
computed to be consistent with grid-mean E3SM output. Interpolation is only used when the input
frequency is equal to or coarser than the frequency of model output. For aircraft measurements, 1-minute
resolution is used to retain high variability and allow matching samples of aerosol and cloud at the same
time. To compare with high-frequency aircraft data, E3SM output is down-scaled to 1-minute resolution
using the nearest grid cell and time slice. The rescale resolution can be adjusted in ESMAC Diags data
preparation code for ESMs running at higher resolution (e.g., kilometer scale grid spacing). All processed
data are saved in a standardized NetCDF format (NETCDF, 2022) and available for downloading (see
data availability section) and direct use.
2.2 Data quality issues and treatments
Many observation datasets used in ESMAC Diags are ARM level-b (quality-controlled) or level-c (value-
added) products, which include quality control (QC) flags to indicate data quality issues. For most
datasets, a QC treatment is applied to remove all data with questionable flags. However, there are certain
datasets or circumstances in which a QC flag is overly strict (too many good data are removed) or not
strict enough (some bad data are not removed). Here we document some of these situations and how we
handle them in our data processing.
2.2.1 ARM Condensation Particle Counter (CPC) measurements
ARM CPC data have several QC values representing failure of different quality checks. One of them
checks if the concentration is greater than a maximum allowable value, which is set to 8,000 cm$^{-3}$ for





model 3010 (CPC, size detection limit 10 nm), 10,000 cm⁻³ for model 3772 (CPCF, size detection limit 10
nm), and 50,000 cm⁻³ for model 3776 (CPCU, size detection limit 3 nm). At SGP, new particle formation
(NPF) events occur frequently when CPC and CPCF measurements can exceed 30,000 cm⁻³. This is much
higher than the maximum allowable value but physically reasonable. Simply removing these large values
results in an underestimation of aerosol number concentration and produces unrealistic diurnal cycle since
they usually occur during the daytime (Tang et al., 2022a). By consulting with the ARM instrument
mentor, we only remove data with critical QC flags, but keep data with this QC flag that is overly
restrictive.
2.2.2 NCAR research flight aerosol number concentration (CN) measurements
NCAR research flight (RF) data used in ESMAC Diags do not include QC flags but occasionally show
suspiciously large or negative aerosol counts. The following minimum and maximum thresholds are
applied to remove suspicious data:
• Total CN from a Condensation Nucleation Counter (CNC, reported as CONCN): minimum = 0,
maximum = 25,000 cm⁻³.
• Total CN from an Ultra-High-Sensitivity Aerosol Spectrometer (UHSAS, reported as
UHSAS100): minimum = 0, maximum = 5,000 cm⁻³.
• Aerosol number size distribution from an UHSAS (reported as CUHSAS_RWOOU or
CUHSAS_LWII): minimum = 0, maximum = 500 cm⁻³ per size bin.
2.2.3 Ship-measured aerosol properties
Aerosol instruments on ships are occasionally contaminated by ship emissions, which present as large
spikes in aerosol and CCN number concentrations. For ARM MARCUS measurements, Humphries
(2020) published reprocessed CN and CCN data to remove ship exhaust contamination using method
described in Humphries et al. (2019). This data is used in this diagnostics package. For MAGIC, we could
not find any ship exhaust contamination information. By visually examining the dataset, a simple
maximum threshold (25,000 cm⁻³ for CPC, 5,000 cm⁻³ for UHSAS100, 2,000 cm⁻³ for CCN at 0.1%
supersaturation and 4,000 cm⁻³ for CCN at 0.5% supersaturation) is applied to remove likely
contamination from ship emissions.
2.2.4 CCN measurements
There are different supersaturation (SS) setting strategies for CCN measurements. Some aircraft
campaigns measured CCN with constant SS (ACE-ENA, HI-SCALE). Some other campaigns measured
CCN with time-varying (scanning) SS (SOCRATES, surface CCN counters at SGP and ENA). However,
the actual SS in a scanning strategy has fluctuations that are different than the target SS. For the latter,
CCN for each SS (0.1%, 0.2%, 0.3% and 0.5%) are obtained by selecting CCN measured within ± 0.05%
of the SS target.
For long-term measurements at SGP and ENA, near-hourly CCN spectra data are available, and a
quadratic polynomial is fit to the spectra such that CCN number concentration can be estimated at any SS
between the measured minimum and maximum SS values. We calculate and output CCN number
concentration from these fits at three target supersaturations (0.1%, 0.2% and 0.5%). The fitted spectra
data provides CCN number concentration at the exact target supersaturations, but the sample number is
slightly smaller due to occasional failure of polynomial fitting.
2.2.5 Contaminated surface aerosol measurements at ENA
The ARM ENA site is located at a local airport. Aerosol measurements at ENA are sometimes
contaminated by aircraft and vehicle emissions, rendering the measurements not representative of the
background environment. Gallo et al. (2020) identified periods when CPC measurements were likely
contaminated from localized emissions (Figure 2a). Their aerosol mask data has 1-min resolution. When
we rescale the data to 1-hr resolution and apply the mask on other coarse time-resolution aerosol
measurements (e.g., ACSM, Figure 2c), we mask hours in which more than half of the hour is flagged by
the aerosol mask. The masking slightly increases the occurrence fraction of small values due to removing
many large values, but it does not change the overall distribution (Figure 2b and 2d). A sensitivity
analysis was performed, showing that 50% is a reasonable threshold to balance removal of contamination
with keeping reasonable data (not shown).

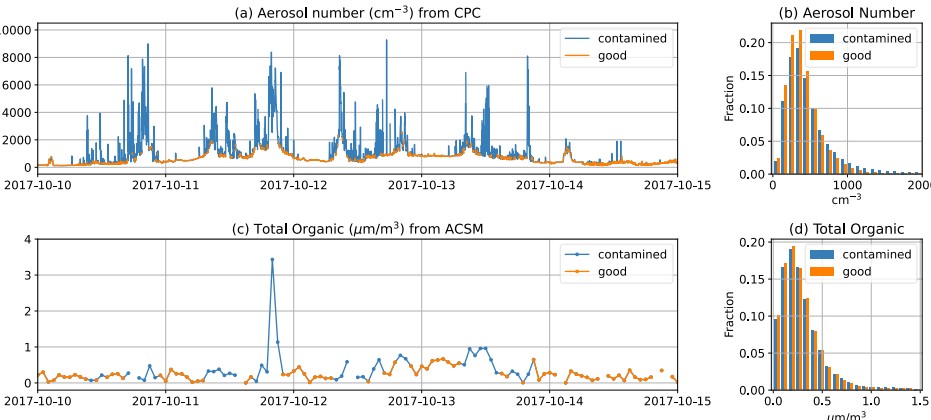


Figure 2: (a) CPC-measured CN from 10 to 15 October 2017 (1-minute resolution) with
local contamination flagged by Gallo et al. (2020). (b) histogram of CPC-measured CN for
all data from 2016-2018. (c) ACSM measured total organic matter from 10 to 15 October
2017 (1-hour resolution). Hours with more than half or the hour flagged in 1-minute CPC
data are masked as contaminated. (d) histogram of ACSM-measured total organic matter
for all data from 2016-2018.
**3. Verification of cloud retrievals with in-situ measurements**
Cloud microphysical properties such as droplet number concentration ($N_d$) and effective radius ($R_{eff}$) are
important variables that connect clouds to other aspects in the climate system such as aerosols and
radiation. Except in field campaigns where in-situ aircraft measurements are available, remote sensing
retrieval algorithms are usually needed to derive these quantities. Several cloud retrieval products from
ground and satellite measurements with different algorithms are used in ESMAC Diags v2. This section



compares these cloud retrievals with in-situ aircraft measurements to assess retrieval limitation,
uncertainty, and utility. $N_d$ and $R_{eff}$ from aircraft measurements taken during HI-SCALE and ACE-ENA
field campaigns are calculated from merged cloud droplet number size distributions (mergedSD) from
three different cloud probes with different size ranges.
Table 3 lists $R_{eff}$ and $N_d$ retrieval products used in ESMAC Diags v2. We retrieved Nd_sat with input
data from VISST products using the algorithms described in Bennartz (2007), but assuming a ratio of the
drop volume mean radius to $R_{eff}$ (commonly referred to as *k*) of 0.74 and a cloud adiabaticity of 80%
(Varble et al., 2023). Other datasets are all available as released products. All retrievals assume a
horizontally homogeneous single-layer liquid phase cloud with constant $N_d$ throughout the cloud layer.
However, retrieval algorithms are usually run for all conditions whenever they return valid values. When
assumptions are not satisfied, retrieved properties may contain large errors and likely alter statistics such
as increasing the occurrence frequency of small $N_d$ as will be shown next.
Table 3: Cloud droplet effective radius $R_{eff}$ and number concentration $N_d$ retrievals

| Variable | Dataset | Platform | Campaign/site | Retrieved from | Reference |
|---|---|---|---|---|---|
| $R_{eff}$ | MFRSRCLDOD | Ground | HI-SCALE, ACE-ENA, SGP, ENA | SW diffuse flux, LWP | (Min and Harrison, 1996; Turner et al., 2021) |
| | VISST | Satellite | HI-SCALE, ACE-ENA, MAGIC, MARCUS, SGP, ENA | Brightness temperature | (Minnis et al., 2011) |
| | Wu_etal | Ground | ACE-ENA, MAGIC, ENA | Radar reflectivity, LWP | (Wu et al., 2020) |
| $N_d$ | Ndrop | Ground | HI-SCALE, ACE-ENA, SGP, ENA | LWP, COD, cloud height | (Riihimaki et al., 2021; Lim et al., 2016) |
| | Nd_sat (calculated from VISST) | Satellite | HI-SCALE, ACE-ENA, MAGIC, MARCUS, SGP, ENA | LWP, COD, CTT | (Bennartz, 2007) |
| | Wu_etal | Ground | ACE-ENA, MAGIC, ENA | Radar reflectivity, LWP | (Wu et al., 2020) |

MFRSRCLDOD: Cloud Optical Properties from the MultiFilter Shadowband Radiometer (MFRSR)
SW: shortwave
COD: cloud optical depth
CTT: cloud top temperature

Figures 3 shows the probability density function (PDF) of $N_d$ retrievals with aircraft measurements for
HI-SCALE and ACE-ENA field campaigns, with the comparison of original temporal resolution versus
30-minute mean, and the use of all available samples and samples that are filtered as overcast (cloud
fraction > 90%) low-level (cloud top height < 4 km) clouds. Figure 4 shows similar plots but for $R_{eff}$.
We also selected two cases with single-layer boundary layer stratus or stratocumulus clouds and plotted
their timeseries of original-resolution and 30-min averaged $R_{eff}$ and $N_d$ in Figure S1. The high-frequency
aircraft measurements and MFRSR/Ndrop retrievals exhibit much larger variability than coarse-frequency
retrievals of Wu_etal and VISST. They frequently sample cloud edges or cloud top/base (for aircraft),



where $N_d$ is typically less than further into the cloud. This causes large occurrence fractions in the lowest
few bins in the $N_d$ PDFs (Figure 3a and 3d). The 30-min VISST products also show large occurrence
fraction in the lowest $N_d$ bin for HI-SCALE (Figure 3a), likely due to high frequency of partial cloudy
condition over continental U.S. Filtering conditions to only include overcast low-level clouds (Figure 3b,
e) and averaging into a coarser resolution (Figure 3c, f) both contribute to the reduction of occurrence
fraction in small-$N_d$ bins, and make the measurements from different instruments more comparable.

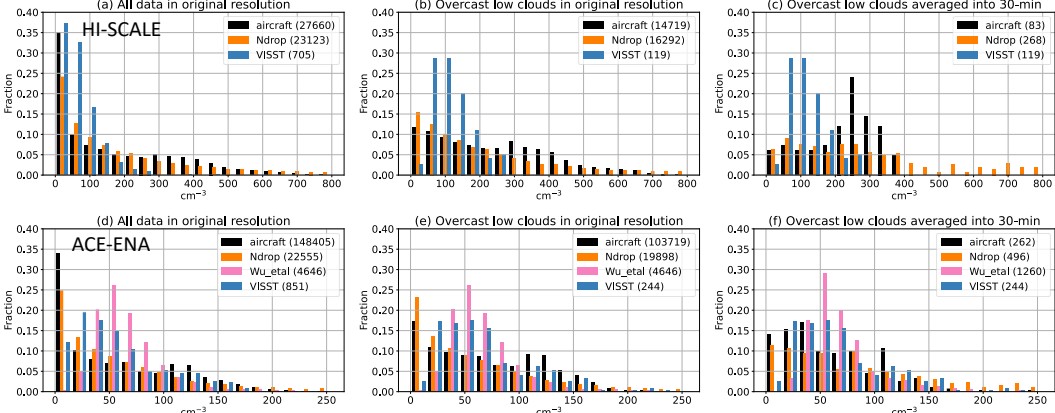


Figure 3: Histogram of $N_d$ from different measurements/retrievals in (top) HI-SCALE and
(bottom) ACE-ENA field campaigns, with total sample numbers in the parentheses. (a) and
(d) use data samples in their original resolution (1 s for aircraft measurements, 20 s for
Ndrop data, 5 min for Wu_etal data, and 30 min for VISST data). (b) and (e) include only
overcast low-cloud situations. For aircraft data, this means $N_d$ is > 1 cm$^{-3}$ for 5 s before
and after the sampling time; for Ndrop and VISST data, it means cloud fraction > 90% and
cloud top height < 4km. (c) and (f) include only overcast low-cloud situations, and
average into 30-min resolution. For all the plots, VISST data with solar zenith angle > 65°
are removed to avoid artifact from sunlight.

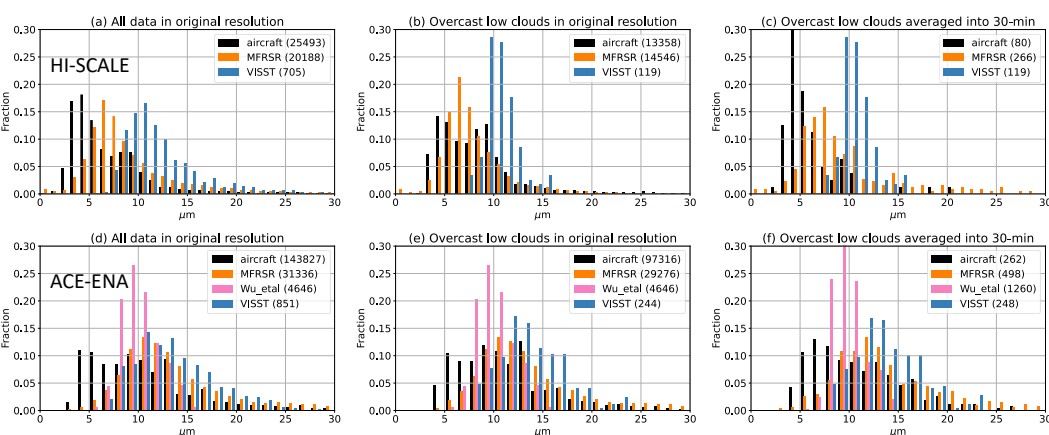




Figure 4: similar as in Figure 3 but for $R_{eff}$.
Overall, the remote sensing retrievals and aircraft measurements produce reasonable ranges of $N_d$ and
$R_{eff}$. Marine clouds (ACE-ENA) have smaller $N_d$ (Figure 3) and larger $R_{eff}$ (Figure 4) than continental
clouds (HI-SCALE). Different retrievals are more consistent with each other for marine clouds than
continental clouds. Different $N_d$ datasets generally agree in mean value, but aircraft and Ndrop data
exhibit broader distributions, likely due to their high sampling frequency that may capture more extreme
conditions with very high or low $N_d$. Moreover, the assumption of a fixed adiabaticity (0.8) in satellite
retrieval will also narrow $N_d$ distribution. For $R_{eff}$, we do not expect different datasets to be perfectly
agree with each other, as cloud droplet size grows with height in the cloud. All remote sensing retrievals
have larger $R_{eff}$ values than aircraft measurements, potentially because remote sensors weight more
towards the upper cloud where droplet size and liquid water content (LWC) are larger. Wu_etal retrieves
vertical profile of $R_{eff}$, and a median value of the $R_{eff}$ profile is used to represent the entire cloud. This
makes Wu_etal retrieval weight less toward large droplets thus its $R_{eff}$ is less than MFRSR and VISST.
VISST data have the largest $R_{eff}$ values, likely because satellite retrievals reflect conditions at the cloud
top. Given the spread in retrieved cloud properties, the limitations and uncertainties of cloud microphysics
retrievals clearly need to be considered when they are used to evaluate model performances.
**4. Structure of diagnostics package**
Figure 5 shows the directory structure of ESMAC Diags v2. It is substantially changed from ESMAC
Diags v1 (Tang et al., 2022a). First, we save all data separately as *raw_data,* which stores all input
datasets collected from field campaigns, and *prep_data*, which stores preprocessed data with standardized
time resolution and quality controls as described in Section 2. The structure is still designed to be flexible
for future extension with additional measurements and/or functionality. Second, the diagnostics functions
now give users more freedom to modify analyses, such as selecting different time periods, performing
additional data filtering or treatments, and examining ACI relationships in specified variable
combinations (for scatter plots, joint histograms or heatmaps). We provide a set of example scripts to
assist users design their own diagnostics based on their needs.



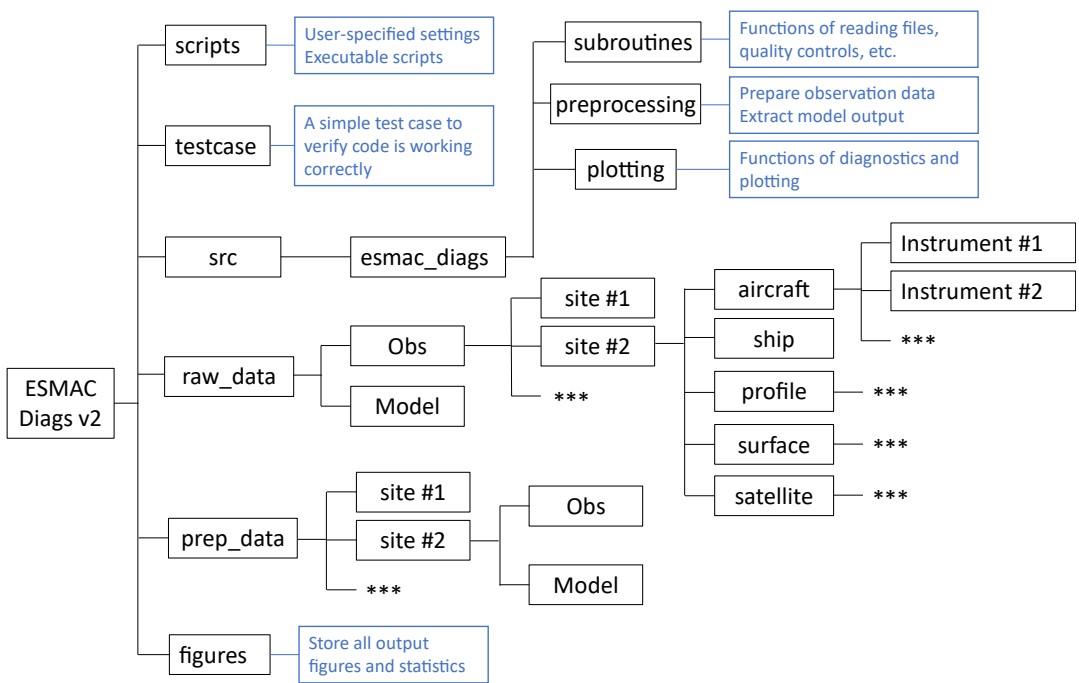

Figure 5: Directory structure of ESMAC Diags v2. Blue boxes describe the functions of the directory. Asterisks represent boxes that follow the same format as those shown in parallel.

ESMAC Diags v1 included diagnostics of aerosol mean statistics (mean, bias, RMSE, correlation), timeseries, diurnal cycle, vertical profiles, mean particle number size distribution, percentiles by height/latitude, and pie/bar charts (Tang et al., 2022a). ESMAC Diags v2 now includes the following new diagnostics that include cloud variables:

- $5^{th}$, $25^{th}$, $50^{th}$, $75^{th}$ and $95^{th}$ percentiles,
- Seasonal cycle at SGP and ENA,
- Histograms for individual variables,
- Scatter plots,
- Joint histograms of two variables, and
- Heatmaps of three variables (mean of one variable binned by two other variables).

The inclusion of two-variable scatter plots, joint histograms, and three-variable heatmaps provides the functionality to study ACI-related relationships. We present a few examples in the next section to demonstrate these new diagnostics.

## 5. Diagnostics Examples

In this section, we show some examples of diagnostics applied to E3SM version 2 (E3SMv2) (Golaz et al., 2022). Compared to the aerosol and cloud parameterizations in E3SMv1 (Rasch et al., 2019; Golaz et



al., 2019), E3SMv2 updated the treatments on dust particles, incorporated recalibration of parameters (Ma
et al., 2022), changed the call order and refactored the code of the Cloud Layers Unified By Binormals
(CLUBB) parameterization, and retuned some parameters (Golaz et al., 2022). We constrain the model
simulations by nudging the horizontal winds towards the 3-hourly Modern-Era Retrospective analysis for
Research and Applications, Version 2 (MERRA-2, Gelaro et al., 2017) with a nudging time scale of 6
hour. Previous studies have shown that with nudging, E3SM can well simulate the large-scale circulations
in reanalyses (Sun et al., 2019; Zhang et al., 2022). The model was run for individual field campaigns
(Table 1) and from 2010 to 2020 for long-term diagnostics at SGP and ENA sites, with hourly model
output saved over the field campaign regions for detail evaluation. As described in Section 2, all
diagnostics for ground and ship campaigns are in 1-hour resolution while diagnostics for aircraft
campaigns are in 1-minute resolution. For aerosol and cloud variables, model raw output variables (not
from instrument simulators) are used in this paper to reveal the intrinsic ACI relationships in E3SM.
However, as can be seen later in this section, instrument simulators can be better used in some diagnostics
to ensure more consistent comparison. Users may choose whether or not to use simulators in their
diagnostics depending on their purpose.

### 5.1. Single-variable diagnostics

Figures 6 and 7 show mean and percentile values of aerosol and cloud properties measured from field
campaigns in the four geographical regions: CUS, ENA, NEP and SO. Figure 6 is for aircraft platforms
and Figure 7 is for ground or ship platforms with satellite data included when available. Note that the
aircraft and ground/ship campaigns may cover different time periods (Table 1), thus some differences
seen between aircraft and ship measurements may be caused by seasonal variation. As cloud
microphysical properties are usually retrieved with assumptions (Section 3), for ground/ship/satellite data,
we only focus on overcast low-level liquid cloud condition here (cloud fraction > 90%, cloud top height <
4 km and ice water path < 0.01 mm). E3SM does not output cloud top height, which is derived using a
weighting integration method as described in Varble et al. (2023).
From both aircraft and ground/ship data, HI-SCALE has much larger aerosol and cloud droplet number
concentrations with smaller droplet sizes compared to other campaigns, which is expected for a
continental environment compared to a marine environment. The cloud optical depth is also greater for
HI-SCALE than other campaigns, which is driven by smaller droplet sizes rather than LWP differences.
Satellite retrievals generally produce smaller $N_d$, LWP, and cloud optical depth with greater $R_{eff}$ than
surface retrievals. As discussed in Section 3, retrieval uncertainties need to be kept in mind when these
retrieved microphysical properties are used to evaluate models.
E3SMv2 overestimates CN (> 10 nm) over CUS, ENA and NEP. Larger particle concentration (CN > 100
nm) is generally underestimated over CUS and overestimated over ENA and NEP. Over SO, E3SMv2
produces fewer small aerosol particles (CN > 10 nm) and about the same number of large aerosol
particles (CN > 100 nm) compared to the observations. These results are confirmed by both aircraft and
ground/ship campaigns, except for the HI-SCALE aircraft campaign where small particles from local
emissions were occasionally observed but unable to be simulated. These results are consistent with our
previous diagnostics for E3SMv1 (Tang et al., 2022a). E3SMv2 also underestimates $N_d$ over CUS and
SO, which corresponds with the underestimation of accumulation mode (> 100 nm) CN over CUS but
underestimation of Aitken mode (> 10 nm) CN over SO. It is possible that over very clean regions such as



SO, small particles are more important in cloud formation than over continental regions such as CUS.
Simulated LWP (LWC) is generally consistent with satellite (aircraft) measurements, but smaller than
ground/ship measurements, which may be partly caused by rain contamination of ground/ship retrievals.
$R_{eff}$ evaluation is less certain given large discrepancies between satellite and ground retrievals.

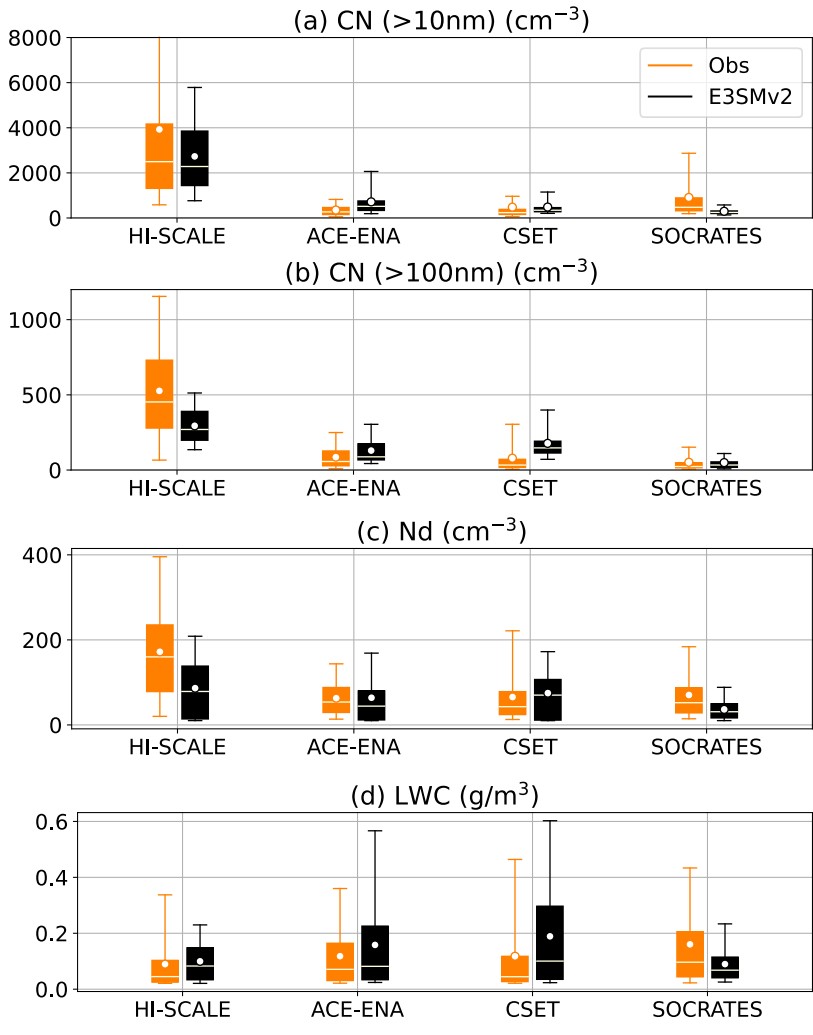


Figure 6: Box-whisker plots of (a) CN for size > 10 nm, (b) CN for size > 100 nm, (c) in-
cloud $N_d$, (d) LWC for all data from aircraft field campaigns at CUS, ENA, NEP and SO
regions from left to right. Boxes denote $25^{th}$ and $75^{th}$ percentiles, whiskers denote $5^{th}$ and
$95^{th}$ percentiles, the white horizontal line represents median values, and the white dot
represents mean values.



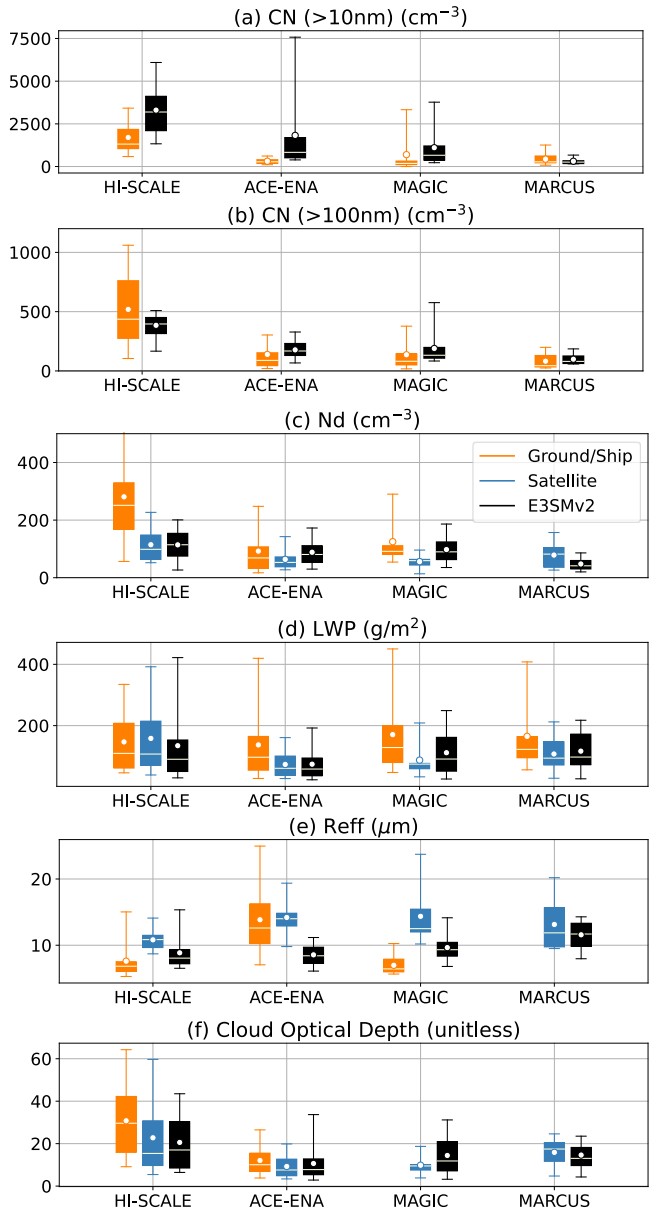

Figure 7: Box-whisker plots of (a) CN for size > 10 nm, (b) CN for size > 100 nm, (c) layer-mean $N_d$, (d) LWP, (e) $R_{eff}$, (f) cloud optical depth for overcast low-level liquid cloud conditions (cloud top height < 4 km, cloud fraction > 90% and ice water path < 0.01 mm) in ground and ship field campaigns at CUS, ENA, NEP and SO regions from left to right. Boxes denote 25$^{th}$ and 75$^{th}$ percentiles, whiskers denote 5$^{th}$ and 95$^{th}$ percentiles, the



white horizontal line represents median values, and the white dot represents mean
values.

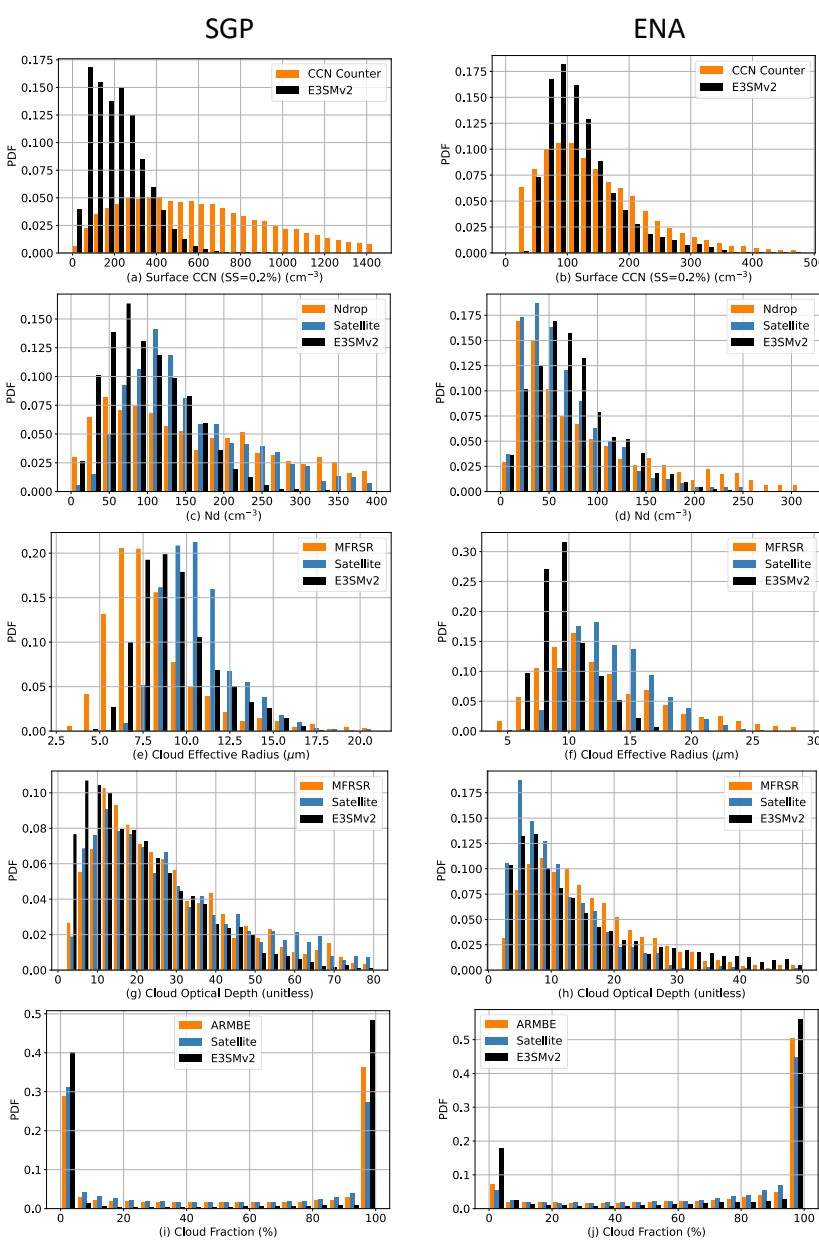


Figure 8: histogram of (from top to bottom) surface CCN number concentration, layer-
mean $N_d$, $R_{eff}$, cloud optical depth and total cloud fraction at (left) SGP from 2011 to
2020 and (right) ENA from 2016 to 2018. Surface CCN and total cloud fraction are using



all-condition samples while $N_d$, $R_{eff}$, cloud optical depth data are filtered for overcast
low-level liquid clouds (cloud top height < 4 km, cloud fraction > 90%, ice water path <
0.01 mm).
Figure 8 shows PDFs of surface CCN number concentration in 0.2% supersaturation, cloud layer mean
$N_d$, $R_{eff}$, cloud optical depth and total cloud fraction for long-term diagnostics at SGP (year 2011-2020)
and ENA (year 2016-2018) sites. E3SMv2 fails to reproduce the long tail of large values in CCN and $N_d$,
especially over SGP. This is consistent with the underestimation of CN (> 100 nm) during the HI-SCALE
field campaign shown in Figures 6 and 7. Compared with ground retrievals, E3SMv2 $R_{eff}$ is larger at
SGP but smaller at ENA. However, satellite-retrieved $R_{eff}$ has larger values than E3SMv2 at SGP. As
discussed before, discrepancies between satellite and ground retrievals can be substantial for some
locations and variables, and considering both in evaluating model performance gives a sense for how
uncertain comparisons are. E3SMv2 generally captures the PDFs of cloud optical depth and total cloud
fraction, although it underestimates the frequency of partial-cloudy conditions and overestimates the
frequency of clear-sky and overcast conditions.

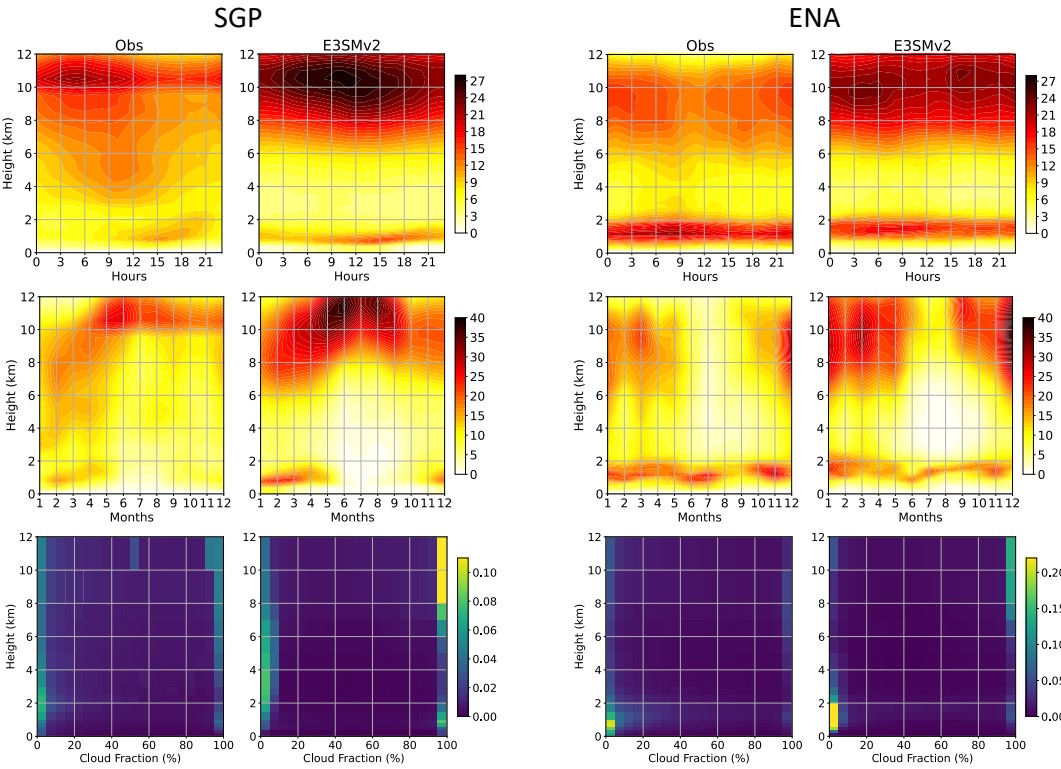


Figure 9: (top) Diurnal cycle, (middle) seasonal cycle, and (bottom) occurrence frequency
of vertical cloud fraction at (left) SGP from 2011 to 2020 and (right) ENA from 2016 to

389     2018.



Figure 9 shows the long-term diagnostics of mean diurnal cycles, seasonal cycles and PDFs of cloud
fraction by height at SGP and ENA sites. At SGP, observations show formation of low clouds in the
afternoon and in late winter through springtime. High clouds peak overnight into the early morning and in
the spring to summer, corresponding to nocturnal deep convective systems common over SGP (Tang et
al., 2022b; Tang et al., 2021; Jiang et al., 2006). These features are reasonably well represented in
E3SMv2, although low-level cloud deepening in the afternoon is not well predicted, and high-level clouds
peak in the late rather than early morning. At ENA, marine stratus or stratocumulus clouds occur in any
month and at any time of the day, but with less frequency in late summer and in afternoon. High clouds
are more frequent in winter months than in summer months and occur throughout the diurnal cycle with a
slight mid-day minimum. These features are well captured by E3SMv2. At both sites, high clouds usually
occur with high fraction (> 95%) while low clouds are more likely associated with small fraction (< 5%)
(bottom row). At SGP, high occurrence of low cloud fraction extends vertically up to the tropopause,
representing frequently occurring deep convection. At ENA, low clouds have less vertical extension but
are more likely to expand to greater fraction. E3SMv2 reproduces these cloud features in occurrence
frequency, with overestimation of occurrence frequency in high (>95%) and low (<5%) cloud fraction
consistent with Figure 8.
Overall, the mean fraction of high clouds is overestimated in E3SMv2. This overestimation has been
reported in many previous studies in the Community Earth System Model (CESM)-E3SM model family
(e.g., Song et al., 2012; Cheng and Xu, 2013; Xu and Cheng, 2013a, b; Tang et al., 2016; Zhang et al.,
2020). However, this is not an apple-to-apple comparison, as cloud fraction in ESMs includes clouds that
are optically very thin that cannot be detected by satellite passive sensors or cloud radar. When satellite
simulators are used, slight underestimation of high cloud fraction by E3SM is seen over most tropical
deep convection regions (Zhang et al., 2019; Xie et al., 2018; Rasch et al., 2019). Unfortunately, our
model does not output cloud vertical profiles from satellite simulators, which prevents a direct apple-to-
apple comparison. Thus, caution should be taken when direct model output is used to compare with
observed cloud fraction.

### 416    5.2. Multi-variable relationships related to ACI

The effective radiative forcing due to ACI processes are complex, nonlinear, and highly uncertain despite
their significant impact on climate. ACI studies are usually conducted by examining relationships
between aerosols, clouds, and radiation variables that are known to interact with one another. Given so
many variable combinations related to ACI, ESMAC Diags v2 provides a framework for users to examine
relationships between the variables they choose with joint histograms, scatter plots and heatmaps. Here
we show a few examples to assess relationships between CCN, $N_d$, LWP, and top of atmosphere (TOA)
albedo.
The dependence of TOA albedo on CCN number concentration for stratiform warm clouds can be
decomposed (e.g., following Quaas et al. (2008)) as:
$$\frac{dA}{dlnCCN} = \left( \frac{\partial A}{\partial lnN_d} + \frac{\partial A}{\partial lnLWP} \frac{dlnLWP}{dlnN_d} \right) \frac{dlnN_d}{dlnCCN} \qquad (1)$$



which allows isolation of "Twomey effect" ($\frac{\partial A}{\partial lnN_d}$) ($\frac{dlnN_d}{dlnCCN}$) and "LWP adjustment" ($\frac{dlnLWP}{dlnN_d}$) associated
with specific ACI processes. Here we use joint histograms and heatmaps to evaluate each component,
$\frac{dlnN_d}{dlnCCN}$, $\frac{dlnLWP}{dlnN_d}$, $\frac{\partial A}{\partial lnN_d}$ and $\frac{\partial A}{\partial lnLWP}$ based on long-term ground and satellite measurements at SGP (2011-
2020) and ENA (2016-2018) sites. The analysis in this section (except Figure 11) is limited to overcast
(cloud fraction > 90%), low-level (cloud top height < 4 km) liquid (ice water path < 0.01 mm) clouds.
Since there is no direct measurement of cloud base CCN concentration from remote sensors, surface CCN
concentration is used in this study and only clouds that are most likely to be affected by surface
conditions are examined. These clouds are identified as having cloud base potential temperature minus
surface potential temperature smaller than 2 K. For satellite measurements, samples with solar zenith
angle greater than 65° are removed to avoid $N_d$ retrieval biases (Grosvenor et al., 2018). The sample
number of (ground, satellite, E3SM) for overcast low-level liquid clouds are (1766, 1217, 6369) at SGP
and (3450, 1345, 2884) at ENA, respectively. To increase sample size for more robust statistics, satellite
retrievals and E3SM outputs over a 5°×5° domain centered on SGP and ENA sites are included. This
increases the sample number to (1766, 71942, 15231) at SGP and (3450, 104260, 28184) at ENA.
Analyses of all-sky conditions and overcast low-level liquid clouds for a single grid point over each site
are shown in Figures S2-S7 in the supplementary material. Increasing sample domain for satellite and
E3SM data does not change the over statistics shown here.
The change of $N_d$ in response to a change of surface CCN number concentration ($\frac{dlnN_d}{dlnCCN}$) is heavily
influenced by processes such as aerosol activation. Figure 10 shows the joint PDFs of $N_d$ and surface
CCN number concentration at 0.1% supersaturation normalized within each CCN bin. Ground and
satellite observations show similar linear fit of $lnN_d - lnCCN$ relation, although ground-based plots have
much smaller sample number. E3SMv2 shows more sensitive $N_d - CCN$ relationships than observations
at both SGP and ENA sites, with the relationship tighter at ENA and more scattered at SGP. As a cross
validation, Figure 11 shows the $N_d - CCN$ relationships from short-term aircraft campaign during HI-
SCALE and ACE-ENA. The comparison with in-situ aircraft measurements confirms that E3SMv2 has
more sensitive $N_d$ to CCN than observations. These results indicate that aerosol activation in E3SMv2
may be too weak in low CCN conditions and too strong in high CCN conditions, which may be related to
the differences in simulated and observed updraft velocity and supersaturation (Varble et al., 2023). Note
that E3SMv2 produces a significant number of small $N_d$ (< 20 cm$^{-3}$) samples (Figure 11). This feature is
reported in Golaz et al. (2022) and is partially removed by setting a minimum threshold of $N_d$ = 10 cm$^{-3}$.
However, as seen in Figure 11, there are still a large number of $N_d$ between 10 and 20 cm$^{-3}$. Further
investigation is underway to diagnose the causes of the abundant low-$N_d$ values. The diagnostics shown
here indicate that a more physical method should be applied to improve the simulated $N_d$.



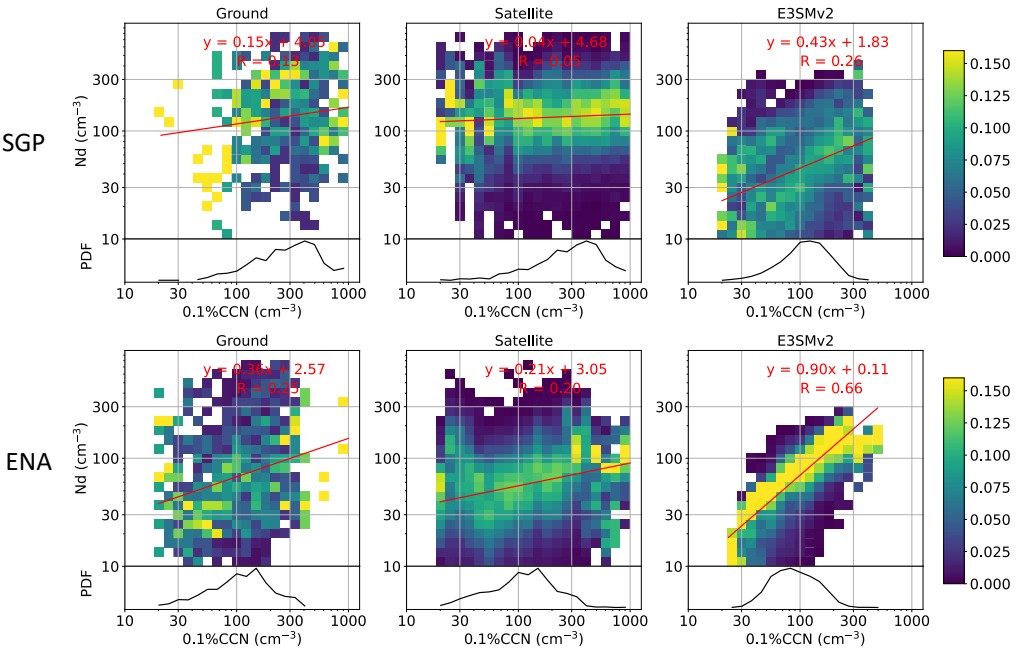

Figure 10: Joint histogram of layer-mean $N_d$ versus surface CCN number concentration at 0.1% supersaturation, normalized within each CCN number concentration bin (PDF of CCN shown in the bottom of each panel). Samples are constrained to likely surface-coupled, overcast low-level liquid clouds (cloud top height < 4 km, cloud fraction > 90%, ice water path < 0.01 mm and potential temperature difference between cloud base and surface < 2 K). Available samples within a 5°×5° region centered on SGP (top) and ENA (bottom) for satellite and E3SMv2 datasets are included. Linear fits and R values are shown in red.



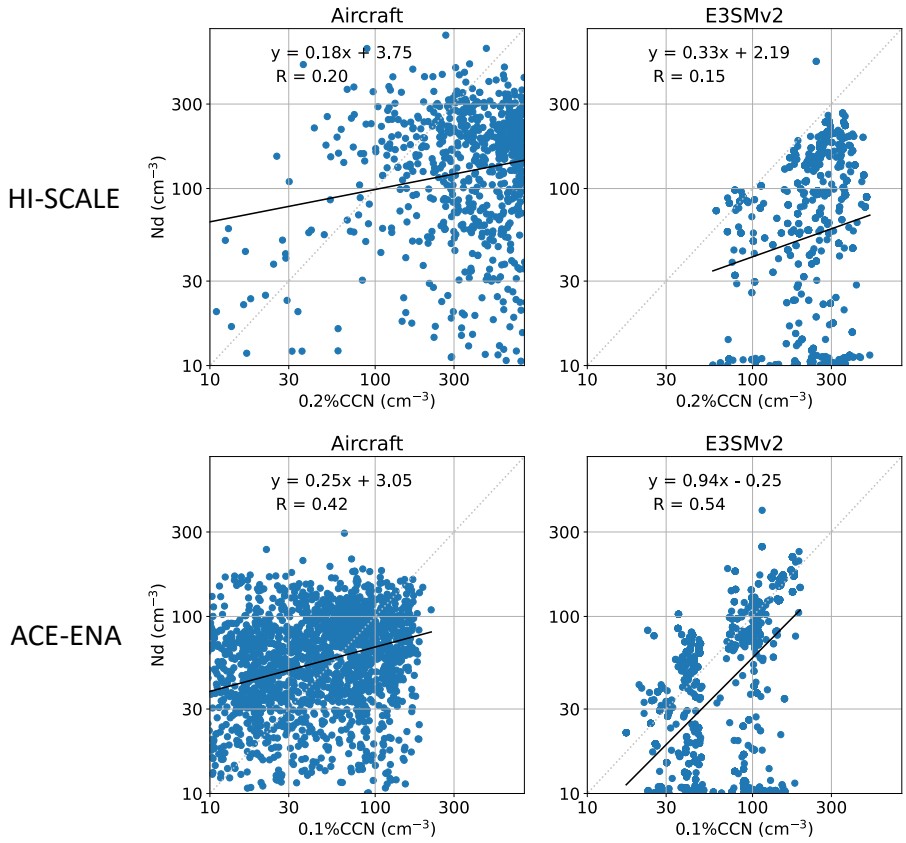

468

Figure 11: Scatter plots for $N_d$ versus CCN along the flight tracks from (top) HI-SCALE and
(bottom) ACE-ENA campaigns. Note that CCN number concentration measurements are
taken under ~0.2% supersaturation for HI-SCALE and under ~0.1% supersaturation for
ACE-ENA. Linear fits and R values are shown in each panel. R = 0.34 (SGP) and 0.74 (ENA)
for E3SMv2 if a minimum $N_d$ = 20 cm$^{-3}$ is applied.

The term $\frac{dlnLWP}{dlnN_d}$ is commonly interpreted as the adjustment of LWP to a perturbation in $N_d$ tied to
suppression of precipitation (increase LWP) or enhancement of evaporation (decrease LWP) (e.g.,
Glassmeier et al., 2019). Gryspeerdt et al. (2019) show that the satellite retrieved LWP over ocean
increases with $N_d$ when $N_d < \sim 30\ cm^{-3}$ and decreases when $N_d > \sim 30\ cm^{-3}$. This relation is also seen
in satellite retrievals at ENA (Figure 12) when using a higher threshold $N_d = 50\ cm^{-3}$ to perform linear
fits (black dashed lines). The linear fit is insignificant for $N_d < 50\ cm^{-3}$ in surface retrievals at both
sites, partly due to small sample number, and also potentially related to drizzle contamination of LWP.
The slope of the LWP – $N_d$ relation in satellite retrievals at SGP is positive for both $N_d$ ranges. This is
opposed to slope shown in the ground retrievals and indicates that retrieval biases may cause opposite



results in ACI studies. The reason why satellite retrievals show positive LWP – $N_d$ relation at SGP is
subject to further investigation.
The E3SMv2 simulated LWP – $N_d$ relation is quite different from satellite retrievals at both sites. At
SGP, it generates a positive slope for $N_d < 50\ cm^{-3}$, and a negative slope for $N_d > 50\ cm^{-3}$. At ENA, it
shows an opposite relation, with LWP decreases for small $N_d$ and increases for large $N_d$. We examined a
few other oceanic regions with frequent stratus or stratocumulus clouds in E3SMv2 and saw similar
behavior (not shown). However, LWP – $N_d$ relation in E3SMv1 performs quite differently, as shown in
Varble et al. (2023). The causes of the different LWP – $N_d$ relation behaviors in E3SM are under further
investigation. Varble et al. (2023) discussed potential physical mechanisms that may affect the different
LWP adjustments in observation and simulation, such as different atmospheric states in E3SM and
observations. Our user-friendly diagnostics package allows these analyses to be routinely performed for
the purpose of better understanding critical model behaviors at process- and mechanistic-levels, providing
observational constraints to facilitate model development efforts.

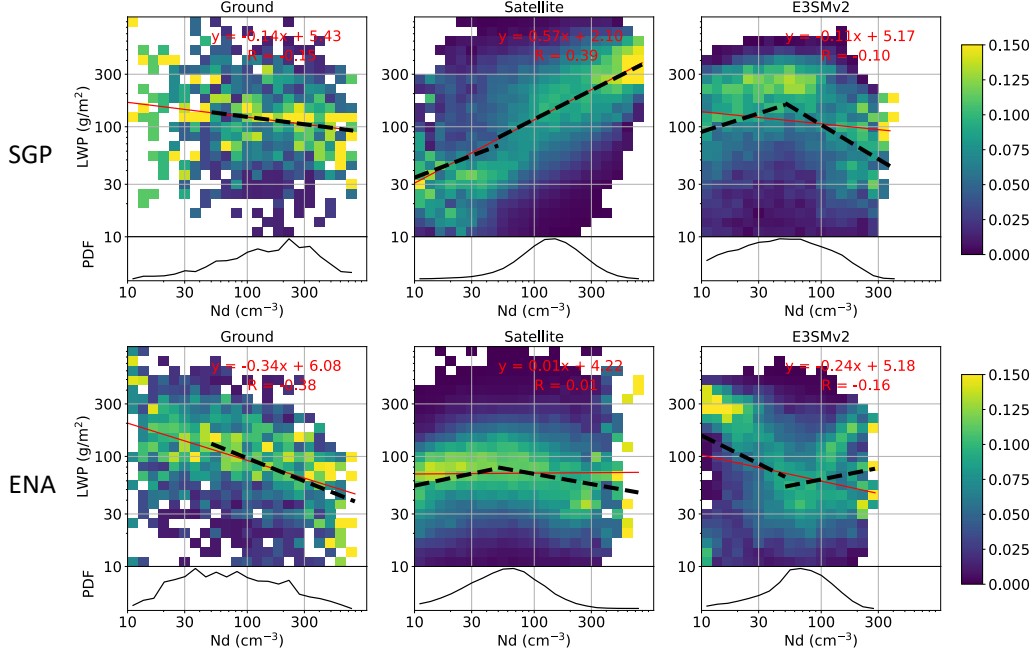


Figure 12: Following Figure 10, but for the $N_d$ bin-normalized joint histogram of LWP
versus $N_d$. Red lines and equations are linear fits for all data samples and black dashed
lines are linear fits for $N_d < 50\ cm^{-3}$ and $N_d > 50\ cm^{-3}$ when the fits are statistically
significant (p < 0.01).



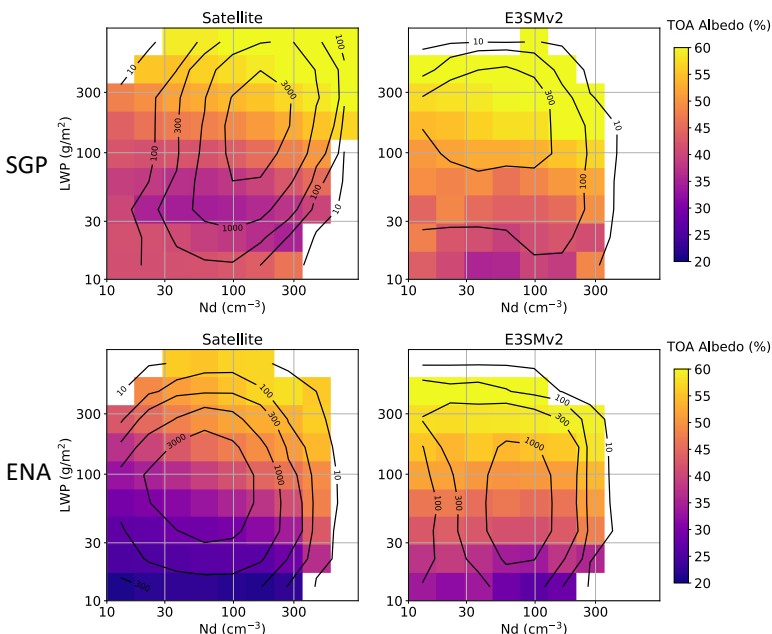

Figure 13: Heatmaps of mean TOA albedo versus LWP and $N_d$ for likely surface-coupled, overcast low-level liquid clouds (cloud top height < 4 km, cloud fraction > 90%, ice water path < 0.01 mm and potential temperature difference between cloud base and surface < 2 K). Data include samples within a 5°×5° region centered on SGP (top) and ENA (bottom). Valid sample number is shown in black contour lines. Grids with valid sample number < 10 are not filled. Ground data is not included, since the TOA albedo is not available.

Figure 13 shows heatmaps of mean TOA albedo with respect to LWP and $N_d$ from which $\frac{\partial A}{\partial lnN_d}$ and $\frac{\partial A}{\partial lnLWP}$ can be derived. At both ENA and SGP, TOA albedo generally increases with increases of LWP and $N_d$, except at SGP when LWP is small. The increasing albedo in small LWP may be due to retrieval artifact as uncertainty becomes large when LWP is small (e.g., < 20 g/m²), solar zenith angle is large (e.g., > 55°), or cloud optical depth is small (e.g., <5) (Grosvenor et al., 2018). TOA albedo at SGP is generally higher than at ENA, which is expected for clouds with smaller droplet sizes. Increasing TOA albedo with increases of LWP is also seen in E3SMv2, but the dependence with $N_d$ is weak. This can be impacted by correlation between solar zenith angle and $N_d$ in E3SM simulation, as discussed in Varble et al. (2023). For a given LWP and $N_d$, TOA albedo is generally higher in E3SMv2 than in satellite observations, indicating that shallow clouds may be too reflective in the model, possibly due to smaller cloud $R_{eff}$ (Figure 8).

The above illustration of single-variable and multi-variable diagnostics present examples to demonstrate the capability of ESMAC Diags v2. More analyses, such as selecting other variables, performing additional data filtering or treatments, and examining ACI relationships with other variable combinations, can be conducted through user-specified settings. A detailed user guide and a collection of example





scripts are included in the diagnostics package to assist users design customized diagnostics suited to their
specific needs.

## 5.  Summary

We developed an Earth System Model aerosol-cloud diagnostics package (ESMAC Diags) to facilitate
routine evaluation of aerosols, clouds and ACI in the U.S. DOE's E3SM model using multiple platforms
of observations. As an updated version of ESMAC Diags v1 (Tang et al., 2022a) which mainly focuses on
aerosol properties, this paper described ESMAC Diags v2 that focuses on both aerosols, clouds, as well as
their interactions. In addition to the short-term field campaigns included in ESMAC Diags v1, long-term
diagnostics from two permanent ARM sites (SGP and ENA, each represents continental and maritime
conditions, respectively) are now conducted to provide more robust evaluation. The newly added multi-
variable joint histograms, scatter plots and heatmaps allow users to examine correlations between
variables that are relevant to the study of ACI.
Ground- and ship-based aerosol measurements are frequently impacted by local-scale emissions sources
such as those from airport or ship exhaust. These local sources are not resolved by coarse-resolution
ESMs, which usually represent an environment averaged within a region of tens to hundreds of kilometers
in size. In ESMAC Diags, we used available contamination-removed aerosol data, such as those from
Gallo et al. (2020) for ENA, and Humphries (2020) for MARCUS, and applied data filtering for other
field campaigns. The observations are harmonized into a uniform data format and temporal resolution that
are comparable with ESMs. Aircraft measurements retain higher resolution (currently 1-min) to preserve
high spatiotemporal variability, although ESMs have to be downscaled for evaluation with aircraft
measurements. This limitation of scale mismatch must be accepted to perform evaluation in current
coarse-resolution ESMs. Nevertheless, as ESM grid spacing approaches a few kilometers via regional
refinement (Tang et al., 2019) or global convection-permitting configuration (Caldwell et al., 2021), the
scale inconsistency between models and observations is reduced. ESMAC Diags can easily adjust the
preprocessing output resolution to facilitate the evaluation of high-resolution model output.
Cloud microphysical properties heavily rely on remote sensing measurements to achieve more robust
sampling, with imperfect retrieval algorithms needed to estimate these variables. Microphysical retrievals
are more uncertain than typical atmospheric state measurements due to the need for many assumptions
related to cloud dynamical and physical processes. We have shown (in Section 3) that ground- and
satellite-based retrievals of $N_d$ and $R_{eff}$ are overall consistent with each other and with in-situ aircraft
measurements, with some systematic differences such as smaller $N_d$ and larger $R_{eff}$ in satellite retrievals.
The discrepancies between different retrievals can be larger for individual days (e.g., Figure S1) but can
be mitigated to some degrees when considering broader statistics (Figures 3 and 4). The usage of multiple
retrieval datasets is critical to understand the robustness of evaluation results, as the spread between
different datasets indicates how robust model-observation differences are and guides interpretations of
model biases to support model development.
Finally, this paper presents a few examples of how well E3SMv2 simulates aerosols, clouds and ACI. We
showed that ESMAC Diags can be used to target further investigation into specific parameterization
components. For example, the analysis of $N_d$ – CCN correlation indicates that E3SMv2 may exhibit too
weak aerosol activation in low CCN conditions and too strong in high CCN conditions; the analysis of
LWP – $N_d$ correlation indicates that either the precipitation suppression and cloud evaporation





mechanisms are not well represented, or there are other mechanisms dominating $LWP - N_d$ correlation in
E3SMv2. These diagnostic analyses provide insights into areas in aerosols, clouds and ACI that warrant
special attention in future model development efforts. As ESMs continuously improve its physical
parameterizations, resolution, and numerical schemes, ESMAC Diags offers a valuable tool for
systematically evaluating the performance of the newer versions of a model in simulating aerosol, clouds
and ACI.



**Code availability**:

*The current version of ESMAC Diags is publicly available through GitHub (https://github.com/eagles-project/ESMAC_diags) under the new BSD license. The exact version (2.1.2) of the code used to produce the results used in this paper is archived on Zenodo (https://doi.org/10.5281/zenodo.7696871). The model simulation used in this paper is version 2.0 (https://doi.org/10.11578/E3SM/dc.20210927.1) of E3SM.*

**Data availability**:

*Measurements from the HI-SCALE, ACE-ENA, MAGIC, and MARCUS campaigns as well as the SGP and ENA sites are supported by the DOE Atmospheric Radiation Measurement (ARM) user facility and available at https://adc.arm.gov/discovery/. Measurements from the CSET and SOCRATES campaigns are supported by National Science Foundation (NSF) and obtained from NCAR Earth Observing Laboratory at https://data.eol.ucar.edu/master_lists/generated/cset/ and https://data.eol.ucar.edu/master_lists/generated/socrates/, respectively. DOI numbers or references of individual datasets are given in Tables S1-S8. All the preprocessed observational and model data used to produce the results used in this paper is archived on Zenodo ( https://doi.org/10.5281/zenodo.7478657).*

**Author contribution**:

*ST, JDF and PM designed the diagnostics package; ST and ACV wrote the code and performed the analysis; PW, XD, FM and MP processed the field campaign datasets and provided discussions on the data quality issues; KZ contributed to the model simulation; JCH contributed to the package design and setup; ST wrote the original manuscript; all authors reviewed and edited the manuscript.*

**Competing interests**:

*Po-Lun Ma is a Topical Editor of Geoscientific Model Development. Other authors declare that they have no conflict of interest.*

*Acknowledgements:*

*This study was supported by the Enabling Aerosol-cloud interactions at GLobal convection-permitting scalES (EAGLES) project (74358), funded by the U.S. Department of Energy, Office of Science, Office of Biological and Environmental Research, Earth System Model Development (ESMD) program area. We thank the numerous instrument mentors for providing the data. This research used resources of the National Energy Research Scientific Computing Center (NERSC), a U.S. Department of Energy Office of Science User Facility operated under Contract No. DE-AC02-05CH11231, using NERSC awards ALCC-ERCAP0016315, BER-ERCAP0015329, BER-ERCAP0018473, and BER-ERCAP0020990. Pacific Northwest National Laboratory (PNNL) is operated for DOE by Battelle Memorial Institute under contract DE-AC05-76RL01830.*



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
