# Peer review of "Earth System Model Aerosol-Cloud Diagnostics Package"

_Geoscientific Model Development, 2023_

## Referee Comment (RC1)

**Review of "Earth System Model Aerosol-Cloud Diagnostics Package (ESMAC Diags) Version 2: Assessments of Aerosols, Clouds and Aerosol-Cloud Interactions Through Field Campaign and Long-Term Observations"**

by Tang et al.,
submitted to Geosci. Model Dev. Discuss.

In this paper, the authors introduce version 2 of the Earth System Model Aerosol-Cloud Diagnostics Packages (ESMAC Diags), which is a tool that enables the comparison of observational datasets with the E3SM (Energy Exascale Earth System Model) model. While version 1 of ESMAC Diags mainly focused on the comparison of aerosol properties between the model and various observations, further capabilities have been added to version 2 to also enable an evaluation of cloud properties and aerosol-cloud interactions (ACI). Besides the technical aspects, which have already been described in its companion paper (Tang et al., 2022), the authors introduce the new capabilities of the revised ESMAC Diags by comparing E3SM to various observational datasets, ranging from in-situ to satellite observations, mainly focusing on clouds and ACI.
The manuscript is well-written and logically structured and fits well into the scope of GMD. The authors nicely outline the new capabilities of ESMAC Diags version 2. The source code of their tool is available to the general public, is accessible and well-documented. Within the analysis package, the authors also provide certain examples of how to preprocess input data and how to run this evaluation tool, enabling new users to quickly familiarize themselves with it, which I highly appreciate. In general, this manuscript merits publication provided that the following comments have been addressed.

General comments:

- The major issue I have regarding this manuscript is one that the authors are already aware of. To ensure a fair comparison between models and observations, such a comparison must be both, scale- and definition-aware. The authors state multiple times that using a satellite simulator would largely improve any comparison between the model and the satellite observations. They also show the example of the overestimated high-level cloud fraction (Fig. 9), which is larger than retrieved by the satellite observations. The authors attribute this bias to different ways cloud fraction is diagnosed in the model and retrieved in the satellite observations. I am wondering how valid any comparison of further cloud properties can be if a comparison already fails at something relatively simple as cloud fraction and how valid any findings with regards to ACI are that are presented in the manuscript. Here, the authors even state that retrieval biases between different observational datasets can strongly affect the analysis of ACI (P21, L483), and I was wondering why they then so confidently compare the model output to the observations if already different observational datasets are not comparable. To this end, the authors need to show that the differences between the native model output and the satellite-simulated model output are at least to some extent comparable to ensure that valid conclusions can be drawn from ESMACS Diags.

- P6, L140-141: Here the authors mention that they downscaled the model data to be in accordance with the aircraft observation. I wonder why they are not going the in opposite direction, namely upscaling the aircraft observation to agree with the model grid size and output frequency. Wouldn't this allow not only for a comparison of mean/median values but would also give information on whether the model values on grid-scale are within the variability of the aircraft observations looking at properties like confidence intervals or similar metrics?
- Following up on the last comment, I was wondering how missing subgrid-scale variability in cloud properties in the model output could potentially affect the comparison between the model and the observations, in particular when the scale of the observations and the model do not match.
- P11, L266-268: The distributions in the histograms look quite different, especially for the HI-SCALE comparison. For that reason, I wonder whether the means actually agree between the respective distribution (I could not find any values in the manuscript)? Anyway, using the mean as a metric for comparison might be not a good idea in my opinion. It is dependent on the underlying distributions (which don't look identical at first glance) and are only comparable when distributions are similar. Either verify that the underlying distributions are similar or replace the mean with a non-parametric estimator like the median, which is not dependent on the underlying distribution. Would the medians still be comparable?
- P14, Fig. 6: It is quite hard to make out differences between E3SM and the observations, in (a) and (b), as CN is almost an order of magnitude higher for HI-SCALE. One could revise this plot by using a log axis or by having a different axis limit for HI-SCALE

Specific comments:

- P2, L58-59: In a recent study (Choudhury and Tesche, 2022), aerosol information from CALIPSO have been used to create a satellite-derived, near-global dataset of CCN. While including it could be a nice addition to a future version of ESMAC Diags, I would at least refer to it in this manuscript to highlight that progress has been made for satellite-derived CCN.
- P3 Fig. 1: Use proper spelling of campaigns (HI-SCALE, ACE-ENA) in the inserts. I would also appreciate if you somewhere state in the plot what the contours are, as not everyone reads the figure caption first.
- P6, L133-134: I assume you rescaled to the output frequency of E3SM. If so, I would write it as such.
- P9, L220-222: The mergedSD dataset combines cloud probes with different size ranges. These size ranges could be different from the size ranges used to derive bulk $N_d$ and $R_{eff}$ in E3SM. Even though I don't expect major differences here, I would appreciate it if the authors could comment on that.
- P9, L237: Are you sure that Fig. 3 shows a PDF? I think it is rather a probability mass function (PMF) as you are using discrete intervals as depicted in the histogram in Fig 3. A PDF is for a continuous distribution, and you only get to probability once you integrate a PDF over a certain size range, which will then give you a PMF.

- P10 Fig. 2/3: Can you add what variables you show in the histograms, i.e. but N_d/R_eff on the x-axis lable.
- P23, L513-514: Here I think you refer to mean TOA albedo, averaged over all albedo bins. You should clarify that in the manuscript.

Technical comments:

- P1, L17: Change *…they are lack of the process-level…* to *…they are lacking process-level…*
- P1, L30: Change *…aerosol-cloud interactions…* to *…ACI…* as it has already been abbreviated.
- P2, L66 and P2, L71: DOE, spell out first before abbreviating.
- P11, L273: Change *profile* to *profiles*.
- P19, L443: Change over to overall.
- P19, L452: … sensitive Nd to CCN relationship …
- P24, L527: … the Earth System Model …

References:

- Choudhury, G. and Tesche, M.: Estimating cloud condensation nuclei concentrations from CALIPSO lidar measurements, Atmos. Meas. Tech., 15, 639–654, https://doi.org/10.5194/amt-15-639-2022, 2022.

---

## Author Comment (AC1)

**Reviewer 1:**

In this paper, the authors introduce version 2 of the Earth System Model Aerosol-Cloud Diagnostics Packages (ESMAC Diags), which is a tool that enables the comparison of observational datasets with the E3SM (Energy Exascale Earth System Model) model. While version 1 of ESMAC Diags mainly focused on the comparison of aerosol properties between the model and various observations, further capabilities have been added to version 2 to also enable an evaluation of cloud properdins and aerosol-cloud interactions (ACI). Besides the technical aspects, which have already been described in its companion paper (Tang et al., 2022), the authors introduce the new capabilities of the revised ESMAC Diags by comparing E3SM to various observational datasets, ranging from in-situ to satellite observations, mainly focusing on clouds and ACI.

The manuscript is well-written and logically structured and fits well into the scope of GMD. The authors nicely outline the new capabilities of ESMAC Diags version 2. The source code of their tool is available to the general public, is accessible and well-documented. Within the analysis package, the authors also provide certain examples of how to preprocess input data and how to run this evaluation tool, enabling new users to quickly familiarize themselves with it, which I highly appreciate. In general, this manuscript merits publication provided that the following comments have been addressed.

We would like to thank the reviewer for taking the time to review this paper and providing helpful comments to improve the paper. The comments are repeated below in black with our reply in blue.

General comments:

• The major issue I have regarding this manuscript is one that the authors are already aware of. To ensure a fair comparison between models and observations, such a comparison must be both, scale- and definition-aware. The authors state multiple times that using a satellite simulator would largely improve any comparison between the model and the satellite observations. They also show the example of the overestimated high-level cloud fraction (Fig. 9), which is larger than retrieved by the satellite observations. The authors attribute this bias to different ways cloud fraction is diagnosed in the model and retrieved in the satellite observations. I am wondering how valid any comparison of further cloud properties can be if a comparison already fails at something relatively simple as cloud fraction and how valid any findings with regards to ACI are that are presented in the manuscript. Here, the authors even state that retrieval biases between different observational datasets can strongly affect the analysis of ACI (P21, L483), and I was wondering why they then so confidently compare the model output to the observations if already different observational datasets are not comparable. To this end, the authors need to show that the differences between the native model output and the satellite-simulated model output are at least to some extent comparable to ensure that valid conclusions can be drawn from ESMACS Diags.

Thank you for the comment. We want to clarify that the current satellite simulator in the Cloud Feedback Model Intercomparison Project (CFMIP) Observation Simulator Package (COSP) only includes

single-layer cloud fraction at high-, mid-, and low- levels and is respected to satellite instruments such as MODIS. It does not have the simulator for cloud vertical profiles or for ground-based radars. Therefore, we are not able to compare ground-based cloud fraction vertical profiles with model simulator output. As some previous studies listed in the manuscript made comparison between native model cloud fraction with radar/lidar observations and made conclusions from that, we want to draw attention of the readers about the limitations of such comparison. We revised the text and move it upfront one paragraph to emphasize this problem (Lines 429-444):

*"Overall, the mean fraction of high clouds looks overestimated in E3SMv2. Similar results has been reported in many previous studies in the Community Earth System Model (CESM)-E3SM model family (e.g., Song et al., 2012; Cheng and Xu, 2013; Xu and Cheng, 2013b, a; Tang et al., 2016; Zhang et al., 2020). However, this is not an apple-to-apple comparison, as cloud fraction in ESMs includes clouds that are optically very thin that cannot be detected by satellite passive sensors or cloud radars. The comparison of high cloud fraction from simulators with corresponding satellite observations showed that E3SM slightly underestimates high clouds over most tropical deep convection regions (Zhang et al., 2019; Xie et al., 2018; Rasch et al., 2019). Unfortunately, ground-based radar simulator of cloud vertical profiles is not available in the current model, which prevents a direct apple-to-apple comparison. Thus, caution should be taken when comparing magnitude of cloud fraction from direct model output and radar measurements. Here we focus on the temporal variabilities (diurnal and seasonal cycles) and the occurrence frequency distribution of cloud fraction, which are less relevant to the detection threshold of cloud radars."*

In ESMAC Diags, we do have applied offline simulators for cloud microphysical properties such as cloud droplet number concentration Nd. For the diagnostics of ACI presented later in this study, we choose to show the native model output as we think that it reveals the "actual" ACI relationship in the model. Whether using native model output or using simulator output can better reveal the model performance of multi-variable relations could be a direction for future exploration but is out of scope of this study. Here our purpose is to show the examples of ACI relation diagnostics ESMAC Diags can do, and ESMAC Diags provides the capability of diagnosing ACI relationship using either native or simulator output. We add the following text to clarify this point (lines 476-479):

*"ESMAC Diags v2 calculate layer-mean $N_d$ from three sources: integrated vertically from native model output, retrieved using Ndrop algorithm and using Nd_sat algorithm, as shown in Table 3. In this study we only show the ACI diagnostics using native model output, as it reveals the "true" ACI relations in the model. Users can choose to use the retrieved $N_d$ in their studies for their purposes."*

• P6, L140-141: Here the authors mention that they downscaled the model data to be in accordance with the aircraft observation. I wonder why they are not going the in opposite direction, namely upscaling the aircraft observation to agree with the model grid size and output frequency. Wouldn't this allow not only for a comparison of mean/median values but would also give information on whether the model values on grid-scale are within the variability of the aircraft observations looking at

properties like confidence intervals or similar metrics?

Aircraft is flying in high speed and change its altitude frequently. In the current model output frequency of 1-hour, it does not make much sense to upscale aircraft measurements into 1-hour resolution as it will cover a long distance and different vertical altitudes. 1-minute is already an upscaled resolution from 1-second aircraft measurements. The resolution can be adjusted in ESMAC Diags, but longer upscale resolution means more measurements need to be discarded due to flight height change or measurement mode change, which will dramatically decrease sample size as aircraft usually only fly a few hours per day. As this work is targeting on the kilometer-scale E3SM version planned in the next few years, we will expect more variabilities in aircraft measurements can be captured in model results. We change the text as following (lines 144-151):

*"For aircraft measurements, 1-minute resolution is used to retain high variability and allow matching samples of aerosol and cloud at the same time. To compare with high-frequency aircraft data, E3SM output is interpolated to the same resolution using the nearest grid cell and time slice. Although the current 1-hour, 1-degree E3SM output could not capture the high variability of the aircraft measurements, we are targeting the exascale E3SM version planned in the next few years. In kilometer scale resolution ESM simulations, the high variability in aircraft measurements will be better captured. In the current diagnostics we only focus on the statistics for the entire campaign."*

• Following up on the last comment, I was wondering how missing subgrid-scale variability in cloud properties in the model output could potentially affect the comparison between the model and the observations, in particular when the scale of the observations and the model do not match.

This is an interesting question when comparing model and observations when the scale does not match, how much variability are contributed by resolved scale that both model and observation can capture and how much are contributed by subgrid scale variability that coarse-resolution model could not capture. From our Fig. 6 it looks that the percentile ranges in high-resolution observations are not always greater than those in coarse-resolution model output. Although the comparison shown here include contributions from both model biases and scale mismatch, it indicates that, at least for simple percentiles, large-scale and low-frequency variabilities dominate over subgrid-scale high-frequency variabilities. We revise the text as following (lines 147-155):

*"Although the current 1-hour, 1-degree E3SM output could not capture the high variability of the aircraft measurements, we are targeting the exascale E3SM version planned in the next few years. In kilometer scale resolution ESM simulations, the high variability in aircraft measurements will be better captured. In the current diagnostics we only focus on the statistics for the entire campaign. As seen later in Section 5.1, coarse-resolution model outputs show similar percentile ranges with the high-resolution aircraft measurements, indicating that for simple percentiles, large-scale variabilities dominate over subgrid variabilities over month-long field campaign periods. Further analysis is needed to understand the importance of other statistics (variance, covariance, etc.) of subgrid scale variabilities."*

• P11, L266-268: The distributions in the histograms look quite different, especially for the HI-SCALE comparison. For that reason, I wonder whether the means actually

agree between the respective distribution (I could not find any values in the manuscript)? Anyway, using the mean as a metric for comparison might be not a good idea in my opinion. It is dependent on the underlying distributions (which don't look identical at first glance) and are only comparable when distributions are similar. Either verify that the underlying distributions are similar or replace the mean with a non-parametric estimator like the median, which is not dependent on the underlying distribution. Would the medians still be comparable?

Thank you for the comment. This sentence is inaccurate as they have quite different mean and median values. We revise this sentence as following to keep the emphasis of the histogram analysis on the distribution (lines 288-291). We also check through the entire manuscript to make sure there is no other misuse of mean or median values:

*"Even after rescaling to the same temporal resolution, aircraft and Ndrop data exhibit broader $N_d$ distributions than satellite retrieval, likely due to their high sampling frequency that may capture more extreme conditions with very high or low $N_d$."*

• P14, Fig. 6: It is quite hard to make out differences between E3SM and the observations, in (a) and (b), as CN is almost an order of magnitude higher for HISCALE. One could revise this plot by using a log axis or by having a different axis limit for HI-SCALE

Thank you for the suggestion. We now separate out aerosol number concentration plots for HI-SCALE in a different y-axis in Fig.6 and Fig.7.

Specific comments:

• P2, L58-59: In a recent study (Choudhury and Tesche, 2022), aerosol information from CALIPSO have been used to create a satellite-derived, near-global dataset of CCN. While including it could be a nice addition to a future version of ESMAC Diags, I would at least refer to it in this manuscript to highlight that progress has been made for satellite-derived CCN.

Thank you for providing the information. We have revised the text as following to add the information and reference (lines 57-64):

*"Satellite remote sensing measurements have global or near global coverage but limited spatial and temporal resolution. They are also facing many challenges to retrieve some variables, especially for aerosol properties such as number concentration, size distribution, chemical composition etc. Some recent studies (e.g., Choudhury and Tesche, 2022) have retrieved cloud condensation nuclei (CCN) number concentration from satellite measurements, which provides a great addition to investigate ACI in global scale. However, large uncertainties exist in satellite retrievals, even for more sophisticated retrieved cloud microphysical properties such as droplet number concentration (e.g., Grosvenor et al., 2018)."*

• P3 Fig. 1: Use proper spelling of campaigns (HI-SCALE, ACE-ENA) in the inserts. I

would also appreciate if you somewhere state in the plot what the contours are, as not everyone reads the figure caption first.

Thank you for the suggestion. We have revised the figure as suggested.

• P6, L133-134: I assume you rescaled to the output frequency of E3SM. If so, I would write it as such.

Thank you for the comment. We revised this sentence to explicitly state the output frequency of E3SM (lines 137-138):

*"Currently, ground, ship and satellite measurements are re-scaled to a 1-hour frequency to be consistent with current E3SM output frequency."*

• P9, L220-222: The mergedSD dataset combines cloud probes with different size ranges. These size ranges could be different from the size ranges used to derive bulk $N_d$ and $R_{eff}$ in E3SM. Even though I don't expect major differences here, I would appreciate it if the authors could comment on that.

Thank you for the comment. We added the following discussion in the text (lines 236-241):

*"The mergedSD covers the size range from 1.5 µm to 9075 µm, covering the entire E3SM cloud droplet size distribution range and extending to rain droplet size range (> 100 µm). For field campaigns used in this study, the aircraft only flied through non-precipitating or drizzling clouds, in which the airborne measurements usually measure rain droplet number 3 to 5 orders of magnitude smaller than cloud droplet number. Therefore, the inclusion of rain droplet size range has ignorable impact on the aircraft-estimated $N_d$ and $R_{eff}$."*

• P9, L237: Are you sure that Fig. 3 shows a PDF? I think it is rather a probability mass function (PMF) as you are using discrete intervals as depicted in the histogram in Fig 3. A PDF is for a continuous distribution, and you only get to probability once you integrate a PDF over a certain size range, which will then give you a PMF.

Thank you for the comment. We have changed the word "PDF" into "occurrence fraction histogram" or "histogram"

• P10 Fig. 2/3: Can you add what variables you show in the histograms, i.e. but $N_d/R_{eff}$ on the x-axis lable.

Thank you for the comment. Figures are revised as suggested.

• P23, L513-514: Here I think you refer to mean TOA albedo, averaged over all albedo bins. You should clarify that in the manuscript.

Thank you for the comment. We revise the sentence as *"In most LWP-$N_d$ bins, TOA albedo at SGP is generally higher than at ENA."* For clarification.

Technical comments:

• P1, L17: Change …they are lack of the process-level… to …they are lacking processlevel…
Revised as suggested.

• P1, L30: Change …aerosol-cloud interac0ons… to …ACI… as it has already been abbreviated.
Revised as suggested.

• P2, L66 and P2, L71: DOE, spell out first before abbreviating.
Revised as suggested.

• P11, L273: Change profile to profiles.
Revised as suggested.

• P19, L443: Change over to overall.
Revised as suggested.

• P19, L452: … sensitive Nd to CCN relationship …
Revised as suggested.

• P24, L527: … the Earth System Model …
Revised as suggested.

References:

• Choudhury, G. and Tesche, M.: Estimating cloud condensation nuclei concentrations from CALIPSO lidar measurements, Atmos. Meas. Tech., 15, 639–654, h1ps://doi.org/10.5194/amt-15-639-2022, 2022.

---

## Author Comment (AC2)

**Reviewer 2:**

General comment:

This paper introduces a climate model diagnosis package that focuses on evaluation of aerosol and cloud related quantities, and applies it to U.S. DOE's E3SM. They find that the recent version of E3SM, v2, reasonably reproduces observed aerosol and cloud properties such as number concentrations of aerosol particles and cloud droplets. They also further investigated potential causes of differences in some simulated quantities comparing to that observed and suggest a direction to improve cloud microphysics schemes.

The aerosol and cloud are known for the biggest uncertainty in the climate modeling thus it is very important to establish standardized methods and comprehensive toolkits for the evaluation, which this study is aiming for and documented about. With that, this paper's scope fits well with that of the GMD journal. The manuscript is also well-written and well-organized. The code and data well prepared for being reused via code sharing (e.g., github) and archiving (i.e., DOI from zenodo), contributing to the open-science effort of the community. I have only a few technical suggestions as listed below and recommend accepting the paper after the suggestions being considered.

We would like to thank the reviewer for taking the time to review this paper and providing helpful comments to improve the paper. The comments are repeated below in black with our reply in blue.

Specific comments:

Figure 1: What is the units for the color shading? Can it be included in the plot for clarity?

Thank you for the comment. We have added the shading label (AOD@550nm) in the plot. The unit of AOD is 1 so it is not added.

Line 142-144: Were the NetCDF files generated following the CF-convention? If so, describing that would be plus.

Thank you for the comment. We currently do not use the CF-convention for the variable names. We will add it in the future update plan of ESMAC Diags.

Figures 6 and 7: Some of whiskers in the box-and-whisker plots are off the chart, I am curious if it would be worth to adjust y-axis range to include the top whiskers. In particular, Figs. 6a and 7c.

Thank you for the comment. Some aerosol variables have long tails in their distribution and adjusting y-axis to include the top whiskers will degrade the visual comparison between model and observations. We added the numbers of the top whiskers in the plot captions if they are out of the chart.

Figures 9, 10, and 12: What are the units for the color shadings? Can it be included in the plot for clarity?

Thank you for the comment. We added the units on the plots.

I wonder if the authors could describe little bit more details about how the model simulation data was prepared and fed to the package. This is important because it could improve applicability of the tool for diverse models beyond E3SM.

Thank you for the comment. We have added the following description in the text (lines 310-313):

*"We also provide the source code of data preparation for observations and model output, and a detailed instruction on how to run the code. Users can revise the code to process their own observational data or model output. All the information is available in the ESMAC Diags github repository."*

For other ESMs, we just need to extract (or calculate) similar variables and save as similar file format to be fed into the package. We have documented it in the README file which is available in the github repository.

---

## Author Response (AR2)

Dear authors,

Thank you for the revised version of your manuscript "Earth System Model Aerosol-Cloud Diagnostics Package (ESMAC Diags) Version 2: Assessments of Aerosols, Clouds and Aerosol-Cloud Interactions Through Field Campaign and Long-Term Observations". Before accepting the manuscript for publication in Geoscientific Model Development, please consider the following reviewer's comment on the revised version:

"The authors addressed most of my concerns. Nevertheless, I am wondering about the validity of the comparison of the Nd-LWP relationships, which do not agree between the three datasets used (ground-based, satellite model). I am wondering how informative such a comparison is if differences are that large and not that well examined, in particular as this is a rather technical paper. If that part would be left out, I think it would not take away anything from the paper."

Thank you,
Axel Lauer
(handling topical editor)

We thank the reviewer for providing valuable feedback to further improve the paper. We understand your concerns regarding the differences of the Nd-LWP relationships between the three datasets, and agree that the large differences are notable and could be seen as a potential limitation. However, we believe that including this comparison, even with the notable differences, is useful for a few reasons:

1. Bring attention to the uncertainties and limitations of the observational datasets. Many previous studies only use satellite retrievals (e. g., Bellouin et al. 2019, Christensen et al. 2022, Quaas et al. 2009, Gryspeerdt et al. 2019, 2020, Hoffman et al. 2020). Including ground-based retrievals, although with much smaller sample size, offers valuable insights of the observation uncertainty as they measure cloud in different directions (from space and from ground) and use different retrieval algorithms for Nd. Our analysis for different sites (Fig. 12) and different sampling strategies (Fig. S3) also indicate the limitation on clouds other than marine overcasting warm clouds. By including these comparisons, we are being transparent about the discrepancies and uncertainties in the data, which is essential in scientific research for understanding the robustness of any results.

2. Reveal the existing biases of LWP susceptibility in ESMs. In contrast to recent observational and LES studies showing a net decrease in LWP in response to a $N_d$ increase, ESMs most commonly produce a net LWP increase (Quaas et al. 2009, Gryspeerdt et al. 2020). In E3SM, we have shown that the overall LWP susceptibility is negative, consistent with the observations. However, the observed inverted V relation of LWP to Nd is oppositely seen in E3SM, indicating possible different mechanisms of LWP susceptibility than in observations. These diagnostical results provide insights for the current ESM biases to guide future model development. This is exactly the goal of developing this diagnostics package.

Once again, we appreciate your feedback and revise the manuscript to highlight the above points. The revised texts are copied below:

Lines 521-534:

*"The slope of the $LWP - N_d$ relation in satellite retrievals at SGP is positive for both $N_d$ ranges. This is opposed to the slope from the ground retrievals and satellite retrievals at ENA. This result reveals a few difficulties on LWP susceptibility studies based on observations. First, limitations of instruments and their platforms (from space or from ground) employed in these observations as well as assumptions and simplifications in their retrieval algorithms, may introduce biases and uncertainties into the retrieved cloud microphysical properties. These biases and uncertainties can be amplified when studying ACI relationships between multiple variables. Second, the robustness of ACI studies is also dependent on geographical locations and cloud types, with environmental dynamic conditions influencing the analytical outcomes. Despite our efforts to constrain meteorology and cloud situations, it is essential to acknowledge the existence of many other factor, such as cloud adiabaticity and solar zenith angle as discussed in Varble et al., 2023), which can impact cloud susceptibility. Given these limitations and uncertainties, researchers should use caution when using observational data to study ACI relationships."*

Line 535-549:

*"The E3SMv2 simulated $LWP - N_d$ relation is quite different from satellite retrievals at both sites. At SGP, it generates a positive slope for $N_d < 50\ cm^{-3}$, and a negative slope for $N_d > 50\ cm^{-3}$. At ENA, it shows an opposite relation, with LWP decreases for small $N_d$ and increases for large $N_d$. The overall LWP susceptibility in E3SMv2 is negative, which is consistent with observations and but differs from most ESMs that produce a positive value (Quaas et al., 2009; Gryspeerdt et al., 2020). However, the observed inverted "V" relation of LWP to Nd is oppositely seen in E3SMv2. We examined a few other oceanic regions with frequent stratus or stratocumulus clouds and saw similar behavior (not shown). This indicates possible different mechanisms of LWP susceptibility in E3SM than in observations. Our user-friendly diagnostics package allows these analyses to be routinely performed for the purpose of better understanding critical model behaviors at process- and mechanistic-levels, providing observational constraints to facilitate model development efforts."*